# The impact of species-wide gene expression variation on *Caenorhabditis elegans* complex traits

Gaotian Zhang [1], Nicole M. Roberto[1], Daehan Lee [1], Steffen R. Hahnel[1] & Erik C. Andersen [1✉]

Phenotypic variation in organism-level traits has been studied in *Caenorhabditis elegans* wild strains, but the impacts of differences in gene expression and the underlying regulatory mechanisms are largely unknown. Here, we use natural variation in gene expression to connect genetic variants to differences in organismal-level traits, including drug and toxicant responses. We perform transcriptomic analyses on 207 genetically distinct *C. elegans* wild strains to study natural regulatory variation of gene expression. Using this massive dataset, we perform genome-wide association mappings to investigate the genetic basis underlying gene expression variation and reveal complex genetic architectures. We find a large collection of hotspots enriched for expression quantitative trait loci across the genome. We further use mediation analysis to understand how gene expression variation could underlie organism-level phenotypic variation for a variety of complex traits. These results reveal the natural diversity in gene expression and possible regulatory mechanisms in this keystone model organism, highlighting the promise of using gene expression variation to understand how phenotypic diversity is generated.

[1] Department of Molecular Biosciences, Northwestern University, Evanston, IL 60208, USA. ✉email: erik.andersen@gmail.com

Quantitative genetic mapping approaches, such as genome-wide association (GWA) and linkage mapping, have been used in a variety of organisms to disentangle the underlying genetic basis of gene expression variation by considering the expression level of each gene as a quantitative trait[1–9]. Expression quantitative trait loci (eQTL) affecting gene expression are often classified into local eQTL (located close to the genes that they influence) and distant eQTL (located farther away from the genes that they influence)[10,11]. Local eQTL are abundant in the genome. For example, over half the genes in yeast and 94.7% of all protein-coding genes in human tissues are hypothesized to have associated local eQTL[7,8]. Genetic variants underlying local eQTL might influence the expression of a specific gene by affecting transcription factor binding sites, chromatin accessibility, other promoter elements, enhancers, or other factors at post-transcriptional levels[12]. Genes encoding diffusible factors, such as transcription factors, chromatin cofactors, and RNAs, are often considered the most likely genes to underlie distant eQTL. Distant eQTL hotspots in several species have been suggested to account for the variation in expression of many genes located throughout the genome[1–3,7,9]. Although many eQTL have been identified in different species, only a few studies have addressed how gene expression variation related to organism-level phenotypic differences[6,8,9,13].

The nematode *Caenorhabditis elegans* is a powerful model to study the genetic basis of natural variation in quantitative traits[14–16]. Genome-wide gene expression variation in different developmental stages and various conditions at the whole-organism or cellular resolution have been discovered and thousands of eQTL have been identified in several studies over the past two decades[3,9,17–23]. However, most of these studies used recombinant inbred lines derived from crosses of the laboratory-adapted reference strain, N2, and the genetically distinct Hawaiian strain, CB4856. Consequently, the observed variation in gene expression and their identified eQTL were limited to the differences among a small number of *C. elegans* strains and only revealed a tiny fraction of the natural diversity of gene expression and regulatory mechanisms in this species. The *C. elegans* Natural Diversity Resource (CeNDR) has a collection of 540 genetically distinct wild *C. elegans* strains[16,24,25]. Variation in organism-level traits has been observed among these wild strains, and many underlying QTL, quantitative trait genes (QTGs), and quantitative trait variants (QTVs) have been identified using GWA mapping studies[15,16]. Therefore, a genome-wide analysis could improve our understanding of the role of gene regulation in shaping organism-level phenotypic diversity, adaptation, and evolution of *C. elegans*.

Here, we investigate the natural variation in gene expression of 207 genetically distinct *C. elegans* wild strains by performing bulk mRNA sequencing on synchronized young adult hermaphrodites. We use GWA mapping to identify 6545 eQTL associated with variation in expression of 5291 transcripts of 4520 genes. We find that local eQTL explain most of the narrow-sense heritability and show larger effects on expression variation than distant eQTL. We identify 46 hotspots that comprise 1828 distant eQTL across the *C. elegans* genome. We further find a wide range of potential regulatory mechanisms that underlie these distant eQTL hotspots. Additionally, we apply mediation analysis to gene expression and other quantitative trait variation data to elucidate putative mechanisms that can play a role in organism-level trait variation. Our results provide a large resource of transcriptome profiles and genome-wide regulatory regions that facilitate future studies. Furthermore, we demonstrate efficient methods to locate causal genes that underlie mechanisms of organism-level trait differences across the *C. elegans* species.

## Results

**Transcriptome profiles of 207 wild *C. elegans* strains**. We obtained 207 wild *C. elegans* strains from CeNDR[25] (Fig. 1a). We grew and harvested synchronized populations of each strain at the young adult stage in independently grown and prepared biological replicates (Fig. 1b). We performed bulk RNA sequencing to measure expression levels and aligned reads to strain-specific transcriptomes (Fig. 1b, Supplementary Fig. 1, and Supplementary Data 1). We focused on protein-coding genes and pseudogenes and filtered out those genes with low and/or rarely detected expression (see "Methods"). Because various hyper-divergent regions with extremely high nucleotide diversity were identified in the genomes of wild *C. elegans* strains[26,27], RNA sequencing reads might be poorly aligned and expression abundances might be underestimated for genes in these regions. For each strain, we filtered out transcripts that fell into the known hyper-divergent regions. We also dropped outlier samples by comparing sample-to-sample expression distances (Supplementary Fig. 1). We harvested animals at the first embryo-laying event rather than at a certain age (hours post-hatching), because we observed variation in ages at the first embryo-laying event across strains. Additionally, we reasoned that expression is influenced primarily by the developmental stage. Here, we evaluated the age of each sample when they were harvested using our expression data and published time-series expression data[28]. We inferred that our animals fit an expected developmental age of 60 to 72 h post hatching (Fig. 1c), during which time the animal is in the young adult stage. The age estimation reflects natural variation in the duration from hatching to the beginning of offspring production for wild *C. elegans* strains. In summary, we obtained reliable expression abundance measurements for 25,849 transcripts from 16,094 genes (15,364 protein-coding genes and 730 pseudogenes) in 561 samples of 207 *C. elegans* strains (Fig. 1b, Supplementary Fig. 1, and Supplementary Data 1), which we used for downstream analyses.

*C. elegans* geographic population structure has been observed previously[24,27,29]. Wild strains from Hawaii and other regions in the Pacific Rim harbor high genetic diversity and group into distinct clusters using genetic relatedness and principal component analysis[24,27,29]. Other strains that were isolated largely from Europe have relatively low genetic diversity because of the recent selective sweeps[24,27,29]. Similar to the previous results, the 207 strains were classified into three major groups consisting of strains from Hawaii, the Pacific coast of the United States, and Europe, respectively, in the genetic relatedness tree (Fig. 1d). Three Hawaiian strains are extremely divergent from all other strains. However, a tree constructed using transcriptome data only exhibited weak geographic relationships and no highly divergent strains (Fig. 1e), suggesting stabilizing selection has constrained variation in gene expression.

**Complex regulatory genetic architectures in wild *C. elegans* strains**. To estimate the association between gene expression differences and genetic variation, we calculated the broad-sense heritability ($H^2$, here calculated as strain-wise variance) and the narrow-sense heritability ($h^2$, here calculated using the SNV matrix in the below GWA mappings) for each of the 25,849 transcript expression traits. We observed a median $H^2$ of 0.31 and a median $h^2$ of 0.06 (Fig. 2a and Supplementary Data 1), indicating strong influences from environmental factors, epistasis, or other stochastic factors on transcript expression variation[7,30,31]. However, $h^2$ of thousands of transcript expression traits indicated a substantial heritable genetic component of the population-wide expression differences.

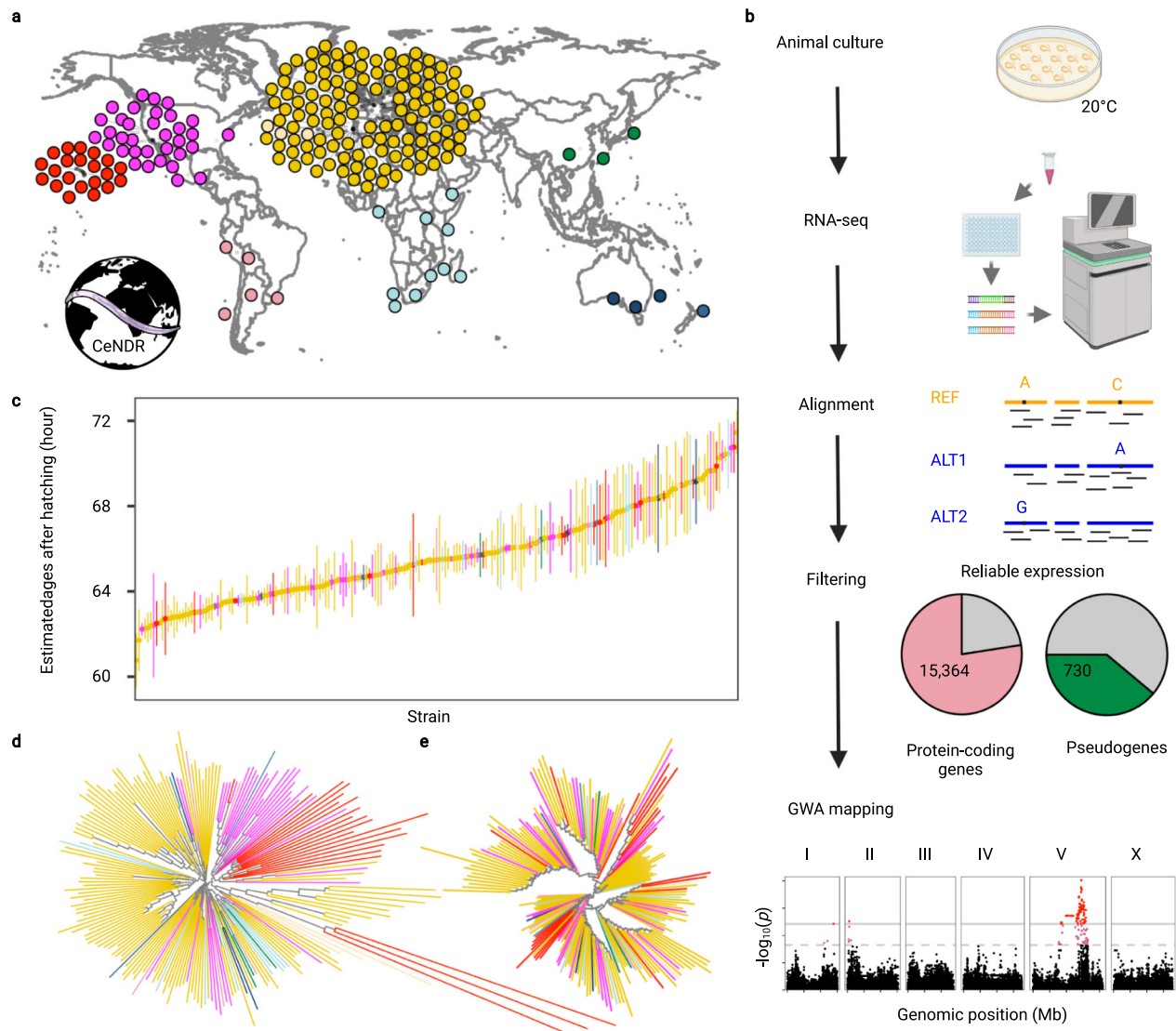

**Fig. 1 Overview of species-wide expression analysis in wild *C. elegans*. a** Global distribution of 205 of the 207 wild *C. elegans* strains that were obtained from CeNDR and used in this study. Strains are colored by their sampling location continent (yellow, Europe; magenta, North America; pink, South America; light blue, Africa; dark blue, Australia; green, Asia), except for Hawaiian strains (red). The two strains missing on the map are lacking sampling locations. **b** Graphic illustration of the workflow to acquire *C. elegans* transcriptome data. Created using BioRender.com. **c** Estimated developmental age (*y*-axis) of 207 wild *C. elegans* strains (*x*-axis), of which 147 strains had three replicates and 60 strains had two replicates. Strains on the *x*-axis are sorted by their mean estimated age from two to three biological replicates. Error bars show the standard deviation of estimated age among replicates of each strain. **d**, **e** Two neighbor-joining trees of the 207 *C. elegans* strains using 598,408 biallelic segregating sites in non-divergent regions (**d**) and expression of 22,268 transcripts in non-divergent regions (**e**) are shown. Strains in **c**–**e** are colored as in **a**.

We performed marker-based GWA mappings to investigate the genetic basis of expression variation in the 25,849 transcripts (Supplementary Data 1). We determined the 5% false discovery rate (FDR) significance threshold for eQTL detection by mapping 40,000 permuted transcript expression traits using the EMMA algorithm[32] and the eigen-decomposition significance (EIGEN) threshold[33] (see "Methods"). In total, we detected 6545 significant eQTL associated with variation in expression of 5291 transcripts from 4520 genes (Fig. 2b and Supplementary Data 2). In close agreement with previous *C. elegans* eQTL studies using recombinant inbred advanced intercross lines (RIAILs) derived from a cross of the N2 and CB4856 strains[3,9], eQTL in this study were mostly found on chromosome arms (61%) relative to centers (33%), which is likely related to the genomic distribution of variation (Table 1). Of the 4520 genes with transcript-level eQTL, we found overrepresentation of nonessential genes (Fisher's exact test, odds ratio:

1.18, *p* value: 0.001) and underrepresentation of essential genes (Fisher's exact test, odds ratio: 0.75, *p* value: 0.001), suggesting stronger selection against expression variation in essential genes than nonessential genes[34]. Gene set enrichment analysis (GSEA) on these 4520 genes showed that proteolysis proteasome-related genes (Fisher's exact test, Bonferroni FDR corrected $p = 3.76\mathrm{E}{-}20$), especially genes encoding E3 ligases containing an F-box domain (Fisher's exact test, Bonferroni FDR corrected $p = 3.73\mathrm{E}{-}15$), are the most significantly enriched class (Supplementary Fig. 2, Supplementary Data 3). Other significantly enriched gene classes include metabolism (Fisher's exact test, Bonferroni FDR corrected $p = 2.92\mathrm{E}{-}12$), stress response (Fisher's exact test, Bonferroni FDR corrected $p = 7.24\mathrm{E}{-}12$), and histones (Fisher's exact test, Bonferroni FDR corrected $p = 3.23\mathrm{E}{-}8$). (Supplementary Fig. 2).

We classified eQTL located within a two-megabase region surrounding each transcript (+/−1 Mb from the transcription

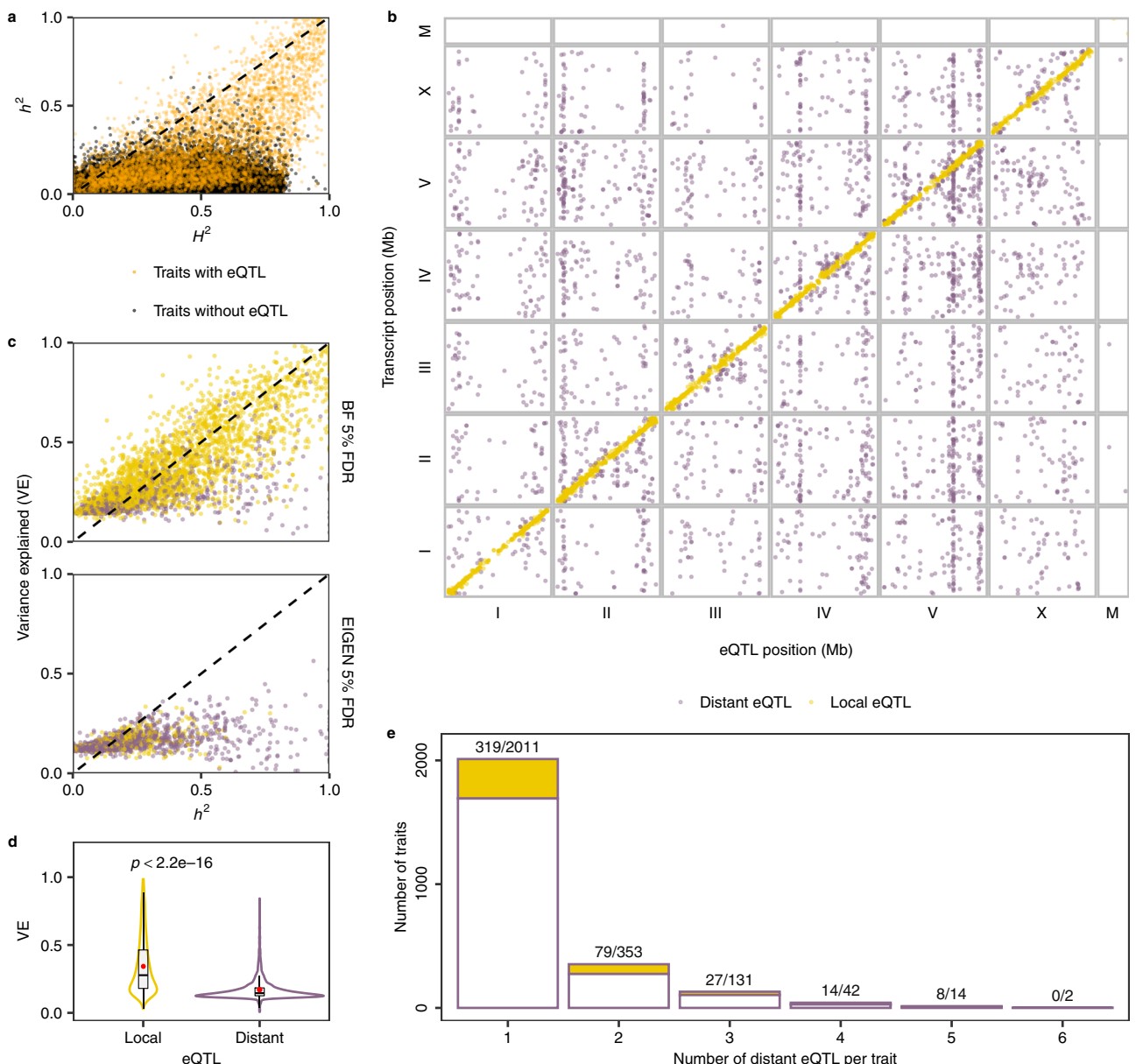

**Fig. 2 Expression QTL map of 207 wild *C. elegans* strains. a** Heritability for 25,849 transcript expression traits with (orange) or without (black) detected eQTL. The narrow-sense heritability ($h^2$, y-axis) for each trait is plotted against the broad-sense heritability ($H^2$, x-axis). **b** The genomic locations of 6,545 eQTL peaks (x-axis) that pass the genome-wide EIGEN 5% FDR threshold are plotted against the genomic locations of the 5,291 transcripts with expression differences (y-axis). Golden points on the diagonal of the map represent local eQTL that colocalize with the transcripts that they influence. Purple points correspond to distant eQTL that are located farther away from the transcripts that they influence. **c** The variance explained (VE) by each detected eQTL (y-axis) that passed Bonferroni (BF) 5% FDR or EIGEN 5% FDR threshold for each trait is plotted against the narrow-sense heritability $h^2$ (x-axis). The dashed lines on the diagonal are shown as visual guides to represent $h^2 = H^2$ (**a**) and $VE = h^2$ (**c**). **d** Comparison of VE between 3185 local and 3,360 distant eQTL shown as Violin plots. The mean and median VE by local or distant eQTL are indicated as red points and horizontal lines in each box, respectively. Box edges denote the 25th and 75th quantiles of the data, and whiskers represent 1.5× the interquartile range. Statistical significance was calculated using a two-sided Wilcoxon test. **e** A histogram showing the number of distant eQTL detected per transcript expression trait. One to six distant eQTL were detected for 2,553 transcript expression traits, of which 447 traits also have one local eQTL. Numbers before slashes (indicated as the golden proportion of each bar) represent the number of traits with a local eQTL in addition to their distant eQTL. Numbers after each slash represent the total number of traits in each category.

start site) as local eQTL and all other eQTL as distant[3,9] (Fig. 2b, Table 1, and Supplementary Data 2). We identified local eQTL for 3185 transcripts from 2655 genes (Fig. 2b, Table 1, and Supplementary Data 2). The 2551 local eQTL that passed the more stringent Bonferroni 5% FDR threshold explained most of the estimated narrow-sense heritability (Fig. 2c). Additionally, we found 3360 distant eQTL for 2553 transcripts from 2382 genes

(Fig. 2b, Table 1, and Supplementary Data 2). Compared to local eQTL, distant eQTL generally explained significantly lower variance (Fig. 2c, d). We found that local eQTL and up to six distant eQTL could jointly regulate the expression of transcripts (Fig. 2e). Because substantial linkage disequilibrium (LD) is observed within ($r^2 > 0.6$) and between ($r^2 > 0.2$) chromosomes in wild *C. elegans* strains[24,27,35], we calculated LD among eQTL of

| Table 1 The distribution of eQTL and SNVs. | | | | | | |
|---|---|---|---|---|---|---|
| **Domain** | **eQTL** | **Local eQTL** | **Distant eQTL** | **Genome** | **Transcripts** | **SNVs** |
| Tip | 388 (5.93%) | 224 (7.03%) | 164 (4.88%) | 7.37 Mb (7.35%) | 1712 (6.62%) | 1628 (7.76%) |
| Arm | 3966 (60.60%) | 2027 (63.64%) | 1939 (57.71%) | 45.89 Mb (45.76%) | 9503 (36.76%) | 12,883 (61.37%) |
| Center | 2183 (33.35%) | 932 (29.26%) | 1251 (37.23%) | 47.01 Mb (46.88%) | 14,622 (56.57%) | 6429 (30.63%) |
| MtDNA | 8 (0.12%) | 2 (0.06%) | 6 (0.18%) | 0.01 Mb (0.01%) | 12 (0.05%) | 51 (0.24%) |
| Total | 6545 | 3185 | 3360 | 100.29 Mb | 25,849 | 20,991 |

Genomic domain coordinates were defined previously[92]. Transcript expression traits and SNVs used for eQTL mappings are listed.

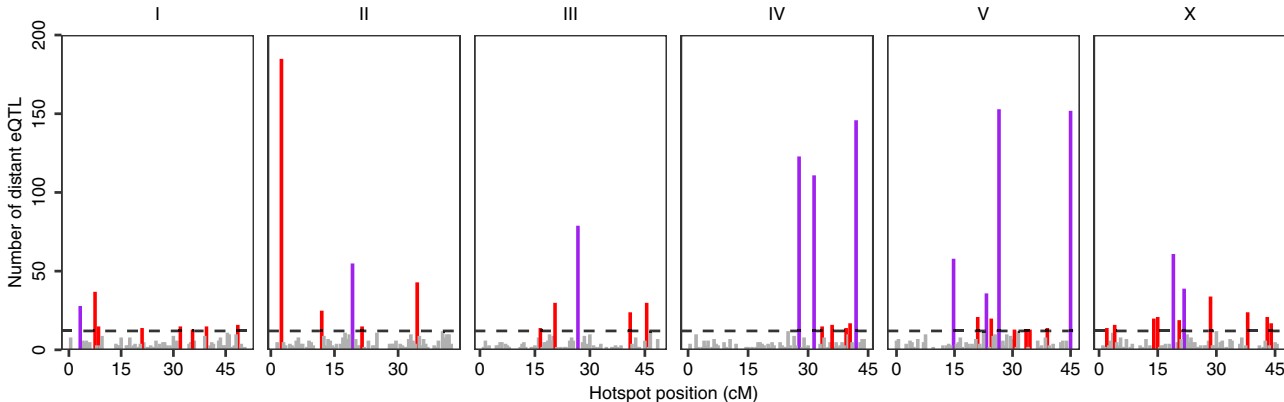

**Fig. 3 Distant eQTL hotspots.** The number of distant eQTL (*y*-axis) in each 0.5 cM bin across the genome (*x*-axis) is shown. The horizontal dashed line indicates the threshold of 12 eQTL. Bins with 12 or more eQTL were identified as hotspots and are colored red. Adjacent bins that all passed the threshold were merged and are colored purple, with the center of the merged bins as the position on the *x*-axis. Bins with fewer than 12 eQTL are colored gray.

each of the 861 transcripts with multiple eQTL. We found low LD among most eQTL, with a median LD of $r^2 = 0.19$ (Supplementary Fig. 3), suggesting complex genetic architectures underlying variation in expression of these transcripts are driven by independent loci.

**A diverse collection of molecular mechanisms underlies distant eQTL hotspots**. Distant eQTL were not uniformly distributed across the genome. Of the 3360 distant eQTL, 1828 were clustered into 46 hotspots, each of which affected the expression of 12–184 transcripts (Fig. 3). GSEA on genes with transcript-level distant eQTL in each hotspot revealed potential shared transcriptional regulatory mechanisms across different genes of the same class in 12 hotspots (Supplementary Fig. 4 and Supplementary Data 3). We further examined the enrichment of genes encoding chromatin cofactors and transcription factors[36–38] in the region of each hotspot and found the hotspot at 30.5-33 cM on chromosome IV was enriched with chromatin cofactor genes (Fisher's exact test, Bonferroni corrected $p = 6$-E5). To suggest if any of these chromatin cofactor genes might be causal, we performed fine mapping on the 110 distant eQTL in this hotspot. We found that a linker histone chromatin cofactor gene, *hil-2*[38], might underlie 33 of the 110 transcripts with distant eQTL in this hotspot. We further performed GSEA for these 33 transcripts and found enrichment in E3 ligases containing an F-box domain (Fisher's exact test, Bonferroni FDR corrected $p = 0.003$) (Supplementary Fig. 5a), heat stress-related genes (Fisher's exact test, Bonferroni FDR corrected $p = 0.01$) (Supplementary Fig. 5b), and transcription factors of the homeodomain class (Fisher's exact test, Bonferroni FDR corrected $p = 0.003$) (Supplementary Fig. 5c). Additionally, we performed fine mapping on distant eQTL in all the other hotspots and filtered for the most likely candidate variants (see Methods for details) (Supplementary Data 4). Then, we focused on the filtered candidate variants that

were mapped for at least four transcript expression traits in each hotspot and are in genes encoding transcription factors or chromatin cofactors. In total, we identified 50 candidate genes encoding transcription factors or chromatin cofactors for 25 hotspots. For example, the gene *ttx-1*, which encodes a transcription factor necessary for thermosensation in the AFD neurons[39,40], might underlie the expression variation of 97 transcripts with distant eQTL in a 1.5 cM hotspot between (44.5-46 cM) on chromosome V. TTX-1 regulates expression of *gcy-8* and *gcy-18* in AFD neurons[39,40], but no eQTL were detected for the two genes likely because we measured the expression of whole animals. Besides the 50 candidate genes, the hundreds of other fine mapping candidates are not as transcription factors or chromatin cofactors, suggesting other mechanisms underlying distant eQTL. Altogether, as previously implicated in other species[7,11,41], our results indicate that a wide range of molecular mechanisms likely cause gene expression variation in *C. elegans*.

**Mediation analysis facilitates candidate gene prioritization**. Mediation analysis seeks to identify the mechanism that underlies the relationship between an exposure (an independent variable) and an outcome (a dependent variable) via the inclusion of one or multiple mediators (intermediary mediating variables). The total effects of the exposure on the outcome include both direct effects that could not be explained by mediators and indirect effects that act through mediators. In quantitative genetics mapping studies, genotypes could affect organism-level phenotypes directly or indirectly through the intermediate effects of gene expression[11]. Therefore, we could use mediation analysis to understand how genetic variants (exposure) affect organism-level phenotypic variation (outcome) through expression variation of one or multiple genes (mediators). We have previously identified mediation effects of *scb-1* expression on responses to several chemotherapeutics and *sqst-5* expression on differential responses to

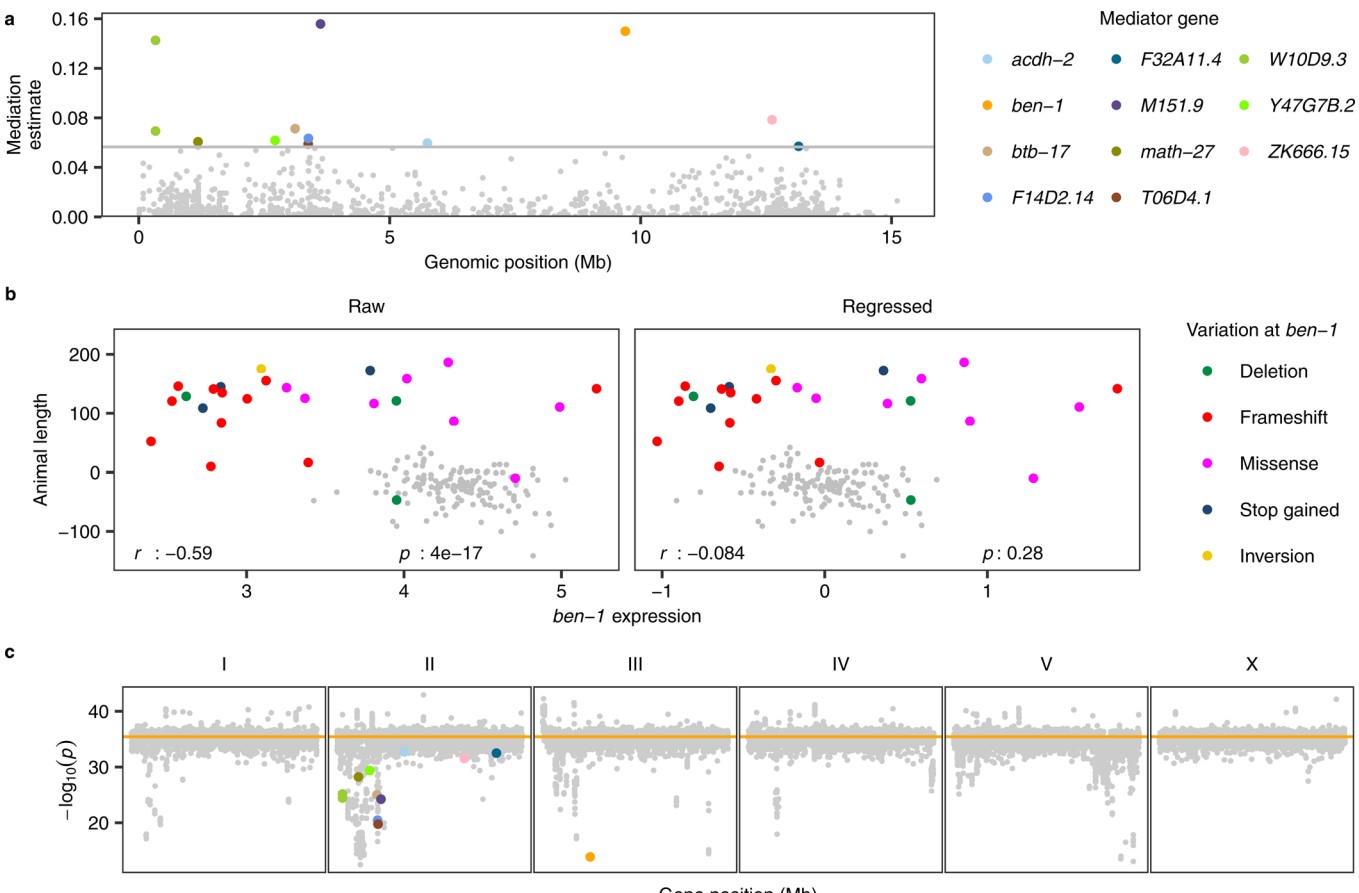

**Fig. 4 Mediation effects of *ben-1* expression on *C. elegans* resistance to albendazole. a** Mediation estimates (*y*-axis) calculated as the indirect effect that differences in expression of each gene play in the overall phenotype are plotted against the genomic position of the eQTL (*x*-axis) on chromosome II. The horizontal gray line represents the 99th percentile of the distribution of mediation estimates. Significant mediators are colored other than gray by their genes as shown in the legend. **b** The correlation of animal length (*y*-axis) to raw *ben-1* expression and to *ben-1* expression regressed by variation at *ben-1* on *y*-axis. The correlation coefficient *r* and the *p* values using the two-sided Pearson's correlation tests were indicated at the bottom. Strains are colored by the type of their genetic variants in *ben-1*. Strains without identified variants are colored gray. **c** Significance at the pseudo variant marker of 25,837 GWA mappings. Each point represents a GWA mapping that is plotted with its −log$_{10}$(*p*) value (*y*-axis) at the pseudo variant marker (III: 3,539,640) against the genomic locations (*x*-axis) of the transcript of which the expression was used in regression for animal length. Points for traits regressed by expression of transcripts identified as significant mediators are colored as in **a**. The orange horizontal line represents the significance of the pseudo variant marker using the raw animal length of 167 strains (Supplementary Fig. 6). GWA mapping results of 12 traits regressed by expression of mitochondrial genes were excluded but all with significance close to the horizontal line. We used the *GWAS()* function in the R package *rrBLUP*[87] to perform the genome-wide mapping with the EMMA algorithm[32] (see "Methods").

exogenous zinc using linkage mapping experiments[9,13]. To validate whether our expression and eQTL data can be used to identify candidate genes, we first performed mediation analysis on one published GWA study of variation in responses to the commonly used anthelmintic albendazole (ABZ)[42].

Previously, wild *C. elegans* strains were exposed to ABZ and measured for effects on development to identify genomic regions that contribute to variation in ABZ resistance. A single-marker GWA mapping was performed first to detect two QTL on chromosomes II and V, but no putative candidate gene was identified. Using a burden mapping approach, prior knowledge of ABZ resistance in parasitic nematodes, and manual curation of raw sequence read alignment files, the gene *ben-1* was found to underlie natural variation in ABZ resistance variation[42]. The single-marker GWA mapping was not able to detect an association between ABZ resistance and *ben-1* variation because of high allelic heterogeneity caused by rare SNVs and structural variants (Supplementary Fig. 6). However, rare SNVs or structural variants might lead to changes in *ben-1* expression and ABZ resistance. We found two distant eQTL, in regions

overlapping with the two organism-level ABZ QTL, for *ben-1* expression variation. Therefore, these results provided an excellent opportunity to test the effectiveness of mediation analysis among organism-level phenotypes, genotype, and gene expression. We performed mediation analysis on the animal length variation (outcome) in response to ABZ, the genetic variation (exposure) at the GWA QTL of the animal length variation, and the expression of 1,157 transcripts (potential mediators) that had eQTL that overlapped with the QTL for the animal length variation. We identified significant mediation effects by the expression of 12 transcripts of 11 genes, including *ben-1* (Fig. 4a). The expression of *ben-1* showed the second-highest mediation effect among the 12 mediators and explained 26% of the total effects via a genetic variation on animal length variation. We found a moderate negative correlation between the expression of *ben-1* and animal length (Fig. 4b), suggesting that expression variation impacts differences in ABZ responses. We further examined genetic variants across strains and found that those strains with relatively low *ben-1* expression and high ABZ resistance all harbor SNVs or structural variants with different

predicted effects (Fig. 4b), suggesting that the extreme allelic heterogeneity at the *ben-1* locus might affect ABZ response variation by reducing the abundance of this beta-tubulin. We have validated the causality of *ben-1* in *C. elegans* response to ABZ using the CRISPR-Cas9 genome editing system for multiple variants in *ben-1*[42–44]. These variants disrupt the function of BEN-1 to affect *C. elegans* responses to ABZ as shown by the developmental stage (animal length) trait[42–44]. We regressed *ben-1* expression by the existence of these genetic variants in *ben-1* and found no correlation between regressed expression and animal length (Fig. 4b), further supporting that the allelic heterogeneity altered *ben-1* expression to affect ABZ response in *C. elegans*. To test the impact of expression variation on phenotypic variation, we regressed animal length by expression of every transcript in our data and performed GWA mappings. Then, we compared the GWA mapping significance value after regression to the original GWA mapping significance value at a pseudo variant marker that represents all the variants in *ben-1* (Fig. 4c and Supplementary Fig. 6)[45]. We found animal length regressed by the expression of *ben-1* showed one of the largest drops in significance, and significance in most of the other mappings was approximately equal to the original significance value (Fig. 4c and Supplementary Fig. 6). These results indicated that increasing *ben-1* expression decreases resistance to ABZ and suggested the applicability of mediation analysis using the expression and eQTL data for other *C. elegans* quantitative traits.

We further applied mediation analysis to another eight previously published studies of *C. elegans* natural variation and GWA mapping studies of different traits, including telomere length[46] (Fig. 5a), responses to arsenic[47] (Fig. 5b), zinc[13] (Fig. 5c), etoposide[48] (Fig. 5d), propionate[49] (Fig. 5e), abamectin[50] (Fig. 5f), dauer formation in response to pheromone[51], and lifetime fecundity[52] (Fig. 5g). Causal variants and genes that partially explained the phenotypic variation in all the eight traits, except for lifetime fecundity, have been identified using fine mappings and genome-editing experiments[13,46–52]. Only one causal gene, *dbt-1* (for arsenic response variation[47]), has eQTL detected and its expression was tested in mediation analysis for arsenic response variation[47] (Fig. 5b). No significant mediation effects were found on arsenic response variation by the expression of *dbt-1*. We also did not observe significant differential expression between strains with different alleles at the previously validated causal *dbt-1* QTV (II:7944817)[47]. Therefore, this causal variant possibly causes arsenic response variation only by affecting enzymatic activity[47] and not the abundance of the *dbt-1* transcript. Instead, we identified *bath-15* as a significant mediator gene for arsenic response variation (Fig. 5b). For the other seven organism-level traits, putative genes whose expression likely mediated the phenotypic variation were detected for six of the traits (Fig. 5). For example, the top mediator gene for the variation in responses to abamectin was *cyn-7*, which is predicted to have peptidyl-prolyl *cis-trans* isomerase activity (Fig. 5f)[53]. For the variation in lifetime fecundity (Fig. 5g), one of the 17 putative mediator genes was *ets-4*, which is known to affect the larval developmental rate, egg-laying rate, and lifespan[54]. To compare with mediation results for lifetime fecundity, we performed fine mapping and identified top candidate genes as described for distant eQTL. We identified 74 candidate genes using fine mapping, without overlapped genes with the 17 mediator genes. Among these 17 mediator genes, seven genes, including *ets-4*, are on different chromosomes from the related QTL, suggesting that mediation analysis nominated new candidate genes that were unable to be detected in fine mappings. Taken together, we concluded that mediation analysis using the newly generated expression and eQTL data facilitates candidate gene prioritization in GWA studies.

## Discussion

*C. elegans* was the first metazoan to have its genome sequenced and has been subjected to numerous genetic screens to identify the genes that underlie different traits, including programmed cell death, drug responses, development, and behaviors. Despite huge efforts by a large research community, over 60% of its genes have not been curated with functional annotations or associated with defined mutant phenotypes[55]. A likely reason is that most *C. elegans* research uses the reference strain N2 under laboratory conditions, and the functions of many genes might only be revealed in natural environments or in different genetic backgrounds[56]. In the last decade, wild *C. elegans* strains have shown phenotypic variation in numerous studies[16,25,29,57–59]. Here, we provide a large resource of transcriptome profiles from wild *C. elegans* strains. Both the raw and processed data are publicly available, and we will further develop an expression browser on CeNDR for easy querying and interactive visualization of expression variation across wild strains. We will also create tools to aid differential expression analysis and visualization between any pair of strains available in our data. We believe our data will facilitate natural variation and evolution research in *C. elegans*.

In addition to generating these data, we used GWA to study gene regulation variation. We detected 6545 eQTL associated with variation in expression of 5291 transcripts of 4520 genes. These genes are enriched in processes, including the proteasome, metabolism, stress response, etc., suggesting gene expression regulation plays an important role in the adaptation of natural *C. elegans* strains to various environments[60,61]. We identified local eQTL that explained most of the narrow-sense heritability ($h^2$) and significantly larger variance than distant eQTL, likely because of higher possibilities of pleiotropy and thus stronger selection pressures. We also observed lower variation in gene expression than in genome sequence and underrepresentation of essential genes among all of the genes identified with eQTL, suggesting stabilizing selection against gene expression as previously observed in *C. elegans* and other species[5,12,62,63].

Although previous *C. elegans* eQTL studies using recombinant inbred lines have revealed rich information on the genetic basis of gene expression variation, mapping using 207 genetically distinct wild strains has the advantage of much greater genetic diversity. We reanalyzed the results of one previous study that used linkage mapping to identify eQTL from the young adult stage of N2xCB4856 recombinant inbred advanced intercross lines (RIAILs)[3,9]. We reclassified 1208 local eQTL and 1179 distant eQTL for 2054 microarray probes of 2003 genes (Supplementary Fig. 7a). Both the eQTL GWA and linkage mappings detected overlapping local eQTL for 454 genes and distant eQTL for 19 genes, indicating that the CB4856 strain carries the common alternative alleles among wild *C. elegans* strains for these 473 loci. However, among the 6545 eQTL that we detected, the strains N2 and CB4856 shared the same genotypes in 4476 eQTL, which could not be discovered using N2xCB4856 recombinant inbred lines. Alternatively, RIAILs might have less linkage disequilibrium between nearby variants and thus smaller eQTL regions of interest than eQTL in wild *C. elegans* strains. The GWA eQTL in this study have a median region of interest of 2.1 Mb (ranged from 12 kb to 18 Mb), whereas the N2xCB4856 RIAILs eQTL showed a median size of 0.55 Mb (ranged from 149 bp to 6.8 Mb), which might make the identification of underlying causal variants easier. The N2xCB4856 RIAILs might also provide greater power than our study, because 1870 eQTL of 1579 genes were only detected using expression data from the N2xCB4856 RIAILs. We further found nine distant eQTL hotspots overlapped between the two studies (Supplementary Fig. 7b). However, these shared hotspots comprise different genes between the two studies, indicating that variation in regulatory

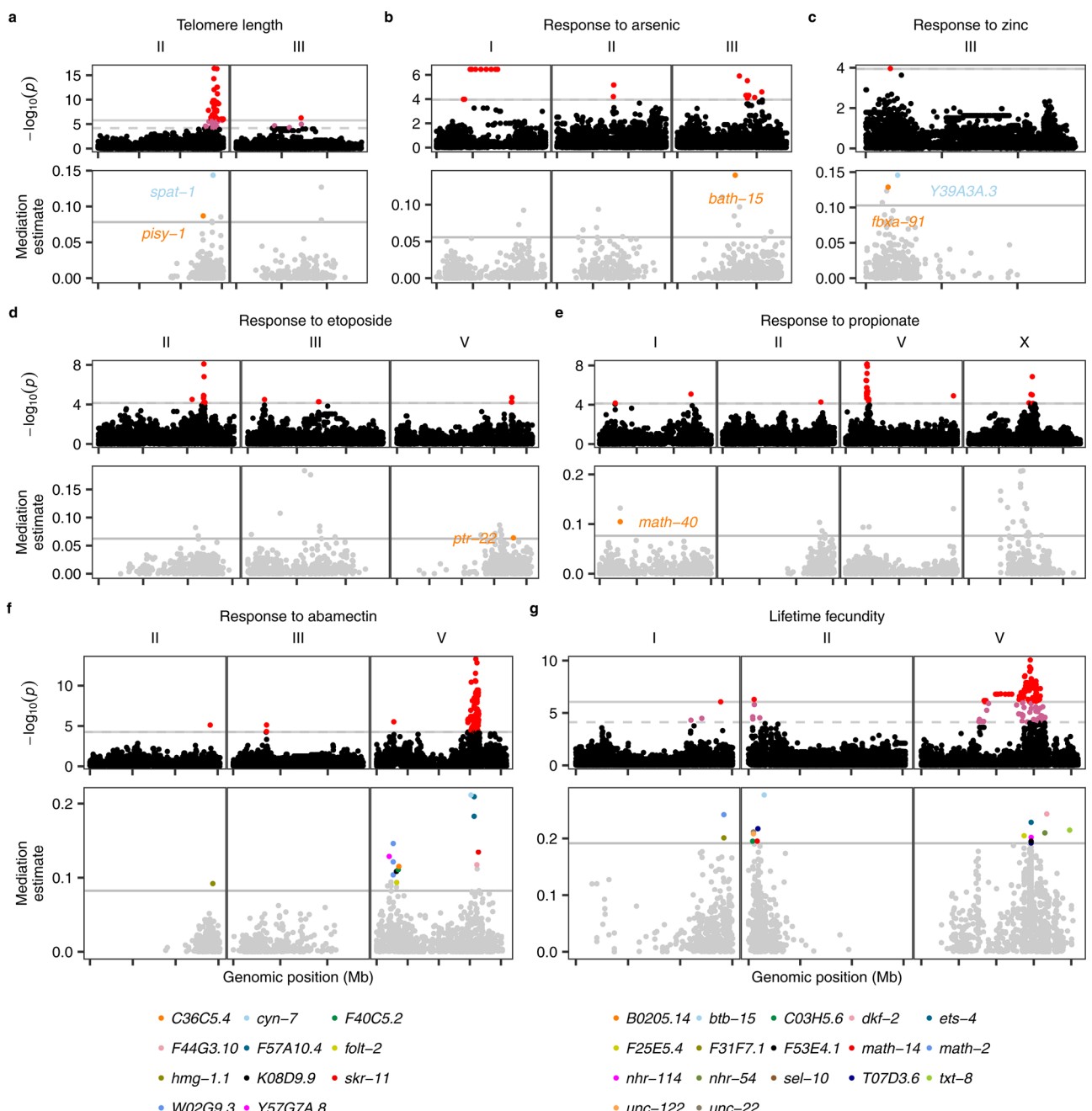

**Fig. 5 Mediation effects of gene expression on variation in seven organism-level phenotypes of *C. elegans*.** GWA mapping and mediation analysis results of natural variation in *C. elegans* telomere length (**a**), responses to arsenic (**b**), zinc (**c**), etoposide (**d**), propionate (**e**), abamectin (**f**), and lifetime fecundity (**g**). Top panel: a Manhattan plot indicating the GWA mapping result for each phenotype is shown. Each point represents an SNV that is plotted with its genomic position (x-axis) against its $-\log_{10}(p)$ value (y-axis) in mapping. SNVs that pass the genome-wide EIGEN threshold (the dotted gray horizontal line) and the genome-wide Bonferroni threshold (the solid gray horizontal line) are colored pink and red, respectively. QTL were identified by the EIGEN (**c**–**f**) or Bonferroni (**a**, **b**, **g**) threshold. Only chromosomes with identified QTL were shown. Bottom panel: mediation estimates (y-axis) calculated as the indirect effect that differences in expression of each gene play in the overall phenotype are plotted against the genomic position (x-axis) of the eQTL. The horizontal gray line represents the 99th percentile of the distribution of mediation estimates. The mediator genes with adjusted $p < 0.05$ and interpretable mediation estimate >the 99th percentile estimates threshold are colored other than gray and labeled in **a**–**e** or below **f**, **g**. Tick marks on x-axes denote every 5 Mb. We used the *GWAS()* function in the R package *rrBLUP*[87] to perform the genome-wide mapping with the EMMA algorithm[32] (see "Methods").

factors is not common between the linkage and association mapping studies. Future research should leverage both types of mapping studies to identify common regulatory mechanisms, focusing on local eQTL.

To further examine whether the eQTL that we found here had been identified previously, we collected eQTL data in *C. elegans* from another seven studies[17–23]. These seven studies measured

the expression of N2xCB4856 RILs and JU1511xJU1926x-JU1931xJU1941 multiparental RILs under different conditions and identified eQTL[17–23]. Our study, the above N2xCB4856 RIAILs eQTL study, and the seven studies represent expression variation under 14 different conditions, such as different environmental conditions and different developmental stages. We examined whether the eQTL that we detected in our study had

been identified in any of the eight studies and the total overlap across all nine studies. We found that 2029 eQTL (31% of the 6545 eQTL) for 1993 transcripts of 1625 genes found in our study were previously detected in at least one other study (Supplementary Fig. 8). The median number of detections of these 2029 eQTL across the 14 conditions of nine studies is three (ranging from two to 14) (Supplementary Fig. 8). The overlapped eQTL identified across studies, especially the 11 eQTL found in all 14 conditions, suggested small effects of developmental stages and environmental conditions on the regulation of these genes.

In addition to the high linkage disequilibrium across the *C. elegans* genome, the recently discovered hyper-divergent genomic regions made this eQTL study challenging. Approximately 20% of the genomes in some wild *C. elegans* strains were found to have extremely high diversity compared to the N2 reference genome[27]. Short-sequence reads of wild *C. elegans* strains often fail to align to the N2 reference genome in these regions and showed lower coverage than in other regions[27]. Similarly, expression levels of genes in hyper-divergent regions could be underestimated because of the poor alignment of RNA-seq reads. Therefore, we only used the expression of transcripts in non-divergent regions to map eQTL and flagged the loci that are in common hyper-divergent regions, where we were less confident in the genotypes of wild strains (Supplementary Data 2). Genes in hyper-divergent regions were enriched in classes that were related to sensory perception, immune response, and xenobiotic stress response[27]. Our data might not capture the full landscape of expression variation in these genes, potentially including some most variable genes, and their local regulatory loci. Furthermore, we only used distant eQTL that are not in common hyper-divergent regions to identify hotspots. Hyper-divergent regions contain poorly characterized SNVs. Therefore, the number of regulatory loci in hyper-divergent regions might be underestimated for both local and distant genes. Future efforts using long-read sequencing are necessary to study the sequence, expression, natural selection, and evolution of genes in hyper-divergent regions, which could improve the understanding on the adaptation of *C. elegans* in various environments.

Variation in gene expression was suggested to impact organism-level phenotypic variation[7,64–66]. Combining previous GWA studies in *C. elegans* with an expression of genes with eQTL, we used mediation analysis to search for organism-level phenotypic variation that can be explained by variation in gene expression. Compared to previous studies using mediation analysis on gene expression and eQTL data from the N2xCB4856 recombinant inbred lines[9,13], we added a multiple testing correction procedures to our mediation analysis. We performed a mediation analysis on ABZ response variation[42]. The causal gene *ben-1* underlying the trait was identified using a burden mapping approach[42] along with prior knowledge[67,68] about the role of beta-tubulin in this drug response. Although two GWA QTL on chromosomes II and V were found, they were identified likely because of their interchromosomal linkage disequilibrium to variants in the *ben-1* locus[42] (Supplementary Fig. 6). The single-marker GWA mapping could not associate ABZ response variation because of the extreme allelic heterogeneity at the *ben-1* locus. However, we used mediation analysis to identify *ben-1* without consideration of prior knowledge or burden mapping results, demonstrating the power of the approach (Fig. 4a). Although the transcriptome is likely to be affected by ABZ treatment, statistical mediation analysis using expression data collected in normal conditions provided evidence on the baseline expression of *ben-1* and other genes that ultimately affected the response of animals to ABZ. The power of this approach is that it does not require the induction of expression differences to cause phenotypic differences (*e.g.*, mediation reflects potential functional differences in levels of genes that vary across

strains in normal growth conditions). Future transcriptome data collected from animals during ABZ treatment could help us to understand how gene expression changes in response to ABZ. We further identified significant mediators for seven other organism-level traits (Fig. 5). The expression of these mediator genes could affect the corresponding phenotypic variation, which should be validated in the future.

Mediation analysis provides an efficient hypothesis-generating approach to be performed in parallel to fine mappings. Additionally, mediator genes could contribute to organism-level phenotypic variation in addition to causal genes identified using fine mappings. One limitation of fine mappings is that searching for causal genes and variants is restricted to the QTL region of interest. Mediation analysis can make statistical connections between the organism-level phenotypes and the expression of genes far away from the QTL. As mentioned above, large GWA QTL regions of interest make it difficult to identify causal genes, which require validation using genome editing. Future *C. elegans* GWA studies should use both fine mappings and mediation analysis to prioritize candidate genes. If the candidate genes overlap between the two approaches, then validation approaches can be initiated using genome editing. In cases where the two approaches identify different candidate genes, prioritization using prior knowledge across all genes identified by both approaches can inform which genes should be tested for validation using genome editing. Previous studies using fine mappings prioritized candidate genes harboring coding variants predicted to have strong functional impacts. In mediation analysis, noncoding variants that likely affect the expression of mediator genes could also be nominated as candidates. For example, upstream variants were suggested to underlie expression variation of the gene *scb-1*, which mediated differences in responses to bleomycin and three other chemotherapeutics[9,69].

The goal of quantitative genetics is to understand the genetic basis and mechanisms underlying phenotypic variation. Here, we showed that mediation analysis, which uses expression and eQTL data to search connections between genetic variants and complex traits, provides additional loci that might further explain phenotypic variation. In the latest version of CeNDR, we have added mediation analysis into the GWA mapping tool[70] using the expression and eQTL data from this study, and provide the results along with other mapping outputs. The framework we developed for mediation analysis complements marker-based GWA mappings and is also applicable to using various other intermediate traits, such as small RNAs, proteins, and metabolites. Any genes and variants underlying variation in these factors can be nominated as candidates for phenotypic validation. Furthermore, we could measure all of these data and complex traits from the exact same samples using *C. elegans*, which can be easily grown at a large scale to have synchronized isogenic populations. Analyses using measurements of mRNAs, small RNAs, proteins, and metabolites could strengthen conclusions about causal genes and mechanisms underlying complex traits using a more holistic perspective of organismal phenotypic variation. We foresee this strategy will greatly improve the powers of quantitative genetic mappings in the future.

## Methods

***C. elegans* strains**. We obtained 207 wild *C. elegans* strains from *C. elegans* Natural Diversity Resource (CeNDR)[25]. Animals were cultured at 20 °C on a modified nematode growth medium (NGMA) containing 1% agar and 0.7% agarose to prevent burrowing and fed *Escherichia coli* strain OP50[71]. Prior to each assay, strains were grown for three generations without starvation or encountering dauer-inducing conditions[71].

**Animal growth and harvest**. We grew and harvested synchronized populations of each strain at the young adult stage with independently grown and prepared biological replicates. Specifically, L4 larval stage hermaphrodites were grown to the

gravid adult stage on 6 cm plates and were bleached to obtain synchronized embryos. Approximately 1000 embryos were grown on each 10 cm plate to the young adult stage and were harvested after the first embryo was observed. M9 solution was used to wash harvested animals twice to remove *E. coli*. Animals were then pelleted by centrifugation (448 × *g* for 1 min) and Trizol reagent (Ambion) was added to maintain RNA integrity before storage at −80 °C.

**RNA extraction.** Frozen samples in Trizol were thawed at room temperature and 100 μL acid-washed sand (Sigma, catalog no. 274739) was added to help to disrupt animal tissues. Then chloroform, isopropanol, and ethanol were used for phase separation, precipitation, and washing steps, respectively. Total RNA pellets were resuspended in nuclease-free water. The concentration of total RNA was determined using the Qubit RNA XR Assay Kit (Invitrogen, catalog no. Q33224). RNA quality was measured using the 2100 Bioanalyzer (Agilent). For the over 600 samples of 207 strains, we performed 30 batches of RNA extraction, with 12–24 samples per batch and replicates of the same strains in different batches. RNA samples with a minimum RNA integrity number (RIN) of 7 were used to construct Illumina sequencing libraries.

**RNA library construction and sequencing.** Illumina RNA-seq libraries were prepared in 96-well plates. Replicates of the same strain were prepared in different 96-well plates. For each sample, mRNA was purified and enriched from 1 μg of total RNA using the NEBNext Poly(A) mRNA Magnetic Isolation Module (New England Biolabs, catalog no. E7490L). RNA fragmentation, first- and second-strand cDNA synthesis, and end-repair processing were performed with the NEBNext Ultra II RNA Library Prep with Sample Purification Beads (New England Biolabs, catalog no. E7775L). The cDNA libraries were adapter-ligated using adapters and unique dual indexes in the NEBNext Multiplex Oligos for Illumina (New England Biolabs, catalog no. E6440, E6442) and amplified using 12 PCR cycles. All procedures were performed according to the manufacturer's protocols. The concentration of each RNA-seq library was determined using the Qubit dsDNA BR Assay Kit (Invitrogen, catalog no. Q32853). Approximately 96 RNA-seq libraries were pooled and quantified with the 2100 Bioanalyzer (Agilent) at Novogene, CA, USA. Each of the pools of libraries was sequenced on a single lane of an Illumina NovaSeq 6000 platform, yielding 150-bp paired-end (PE150) reads.

In total, RNA-seq data of 608 samples from 207 wild *C. elegans* strains in seven pooled libraries were obtained with an average of 32.6 million reads per sample and a minimum of 16.6 million reads. Of the 207 strains, 194 strains with three replicates, and 13 strains with two replicates.

**Sequence processing and expression abundance quantification.** Adapter sequences and low-quality reads in raw sequencing data were removed using *fastp* (v0.20.0)[72]. *FastQC* (v0.11.8) analysis (http://www.bioinformatics.babraham.ac.uk/projects/fastqc) was performed on trimmed FASTQ files to assess read quality (adapter content, read-length distribution, per read GC content, etc.). For RNA-seq mapping, SNV-substituted reference transcriptomes for each of the wild *C. elegans* strains were generated using *BCFtools* (v.1.9)[73], *gffread* (v0.11.6)[74], the N2 reference genome (WS276), a GTF file (WS276)[53], and the hard-filtered isotype variant call format (VCF) 20200815 CeNDR release (Supplementary Fig. 1). Transposable element (TE) consensus sequences of *C. elegans* were also extracted from Dfam (release 3.3)[75] using scripts (https://github.com/fansalon/TEconsensus). We used *Kallisto* (v0.44.0) to (1) pseudoalign trimmed RNA-seq reads from each sample to the transcriptome index built from the strain-specific SNV-substituted reference transcriptome (65,173 transcripts) and TE consensus sequences (157 TEs) and (2) quantify expression abundance at the transcript level[76]. On average, 31.3 million reads pseudoaligned to the transcriptome index per sample with a minimum of 15.5 million reads, which were sufficient to capture the expression of more than 70% of the *C. elegans* reference genome genes. We used the 608 samples of 207 strains and 39,008 transcripts of protein-coding genes and pseudogenes in our analysis.

**Selection of reliably expressed transcripts.** We first normalized the raw counts of transcript expression abundances without the default filtering of low abundance transcripts using the R package *sleuth* (v0.30.0)[77]. Then, we filtered 26,043 reliably expressed transcripts of 16,238 genes by requiring at least five normalized counts in all the replicates of at least ten strains (Supplementary Fig. 1). We also removed data for transcripts that were in the hyper-divergent regions on a per strain basis, because hyper-divergent regions varied across *C. elegans* strains[27]. The data of 3775 transcripts were removed in at least one strain. After this filtering, the total number of strains that retained data in the 3775 transcripts was lower than 207. To maintain relatively high power in eQTL mappings, we required that at least 100 strains should have retained data in each transcript. So we thoroughly filtered out 194 of the 3775 transcripts that did not meet this final requirement. In summary, we collected reliable expression abundance for 25,849 transcripts of 16,094 genes (15,364 protein-coding genes and 730 pseudogenes).

**Selection of well-clustered samples.** We used sample-to-sample distance to select well-clustered samples (Supplementary Fig. 1). We first summarized raw counts of reliably expressed transcripts into gene-level abundances using the R package *tximport* (v1.10.1)[78]. Then, we performed variance stabilizing transformations on

the gene expression profile using the *vst()* function in the R package *DESeq2* (v1.26.0), which generated log₂ scale normalized expression data[79]. Sample-to-sample pairwise Euclidean distances among the 608 samples were calculated using the generic function *dist()* in R (v3.6.0)[80]. Our basic assumption is that intra-strain distances among replicates should be smaller than inter-strain distances. Because the majority of the 207 wild *C. elegans* strains exhibit low overall genetic diversity (Fig. 1d)[24,35,81], we required that the intra-strain distances of replicates be smaller than the median of inter-strain distances of the strain to other strains. Specifically, for each strain, if all of its intra-strain distances were smaller than the median of its inter-strain distances, then all of its replicates were kept. If none of its intra-strain distances were smaller than the median of its inter-strain distances, then all samples of the strain were removed. For strains with three replicates, if one or two of its three intra-strain distances were smaller than the median of its inter-strain distances, then the two replicates with the minimum distances were kept. After the removal of some outlier samples, the median of inter-strain distances would change. Therefore, we repeatedly performed the procedures of data transformation, sample-to-sample distance calculation, and filtering by comparing inter- and intra-strain distances until no more samples were removed. Eventually, 561 samples of 207 strains were selected as well-clustered samples, which comprised 147 strains with three replicates and 60 strains with two replicates.

**Transcript expression abundance normalization.** We used the function *norm_factors()* in the R package *sleuth* (v0.30.0)[77] to compute the normalization factors for each sample using the raw transcripts per million reads (TPM) of 22,268 reliably expressed transcripts in non-divergent regions of the 207 strains and their well-clustered samples. Then, we normalized the raw TPM of all the 25,849 reliably expressed transcripts of each sample with the normalization factors and used log₂(normalized TPM + 0.5) for downstream analysis unless indicated otherwise.

**Sample age estimation.** To further verify the homogeneous developmental stage of our samples, we evaluated the age of each sample when they were harvested using the R package *RAPToR* (v1.1.3)[28] (Supplementary Fig. 1). As the requirement of the package, we first generated gene-level expression abundances. Raw TPM of 22,268 reliably expressed transcripts in non-divergent regions were summarized into abundances of 13,637 genes using the R package *tximport* (v1.10.1)[78]. Normalization factors for each sample using gene-level abundances were calculated as described for transcript level and were used to normalize gene-level TPM. Correlation of log₂(normalized TPM + 0.5) of our data against the reference gene expression time series (Cel_YA_2) in *RAPToR* was computed using the function *ae()* in *RAPToR* with 10,489 intersected genes and default parameters.

**Genetic and expression relatedness.** Genetic variation data for 207 *C. elegans* isotypes were acquired from the hard-filtered isotype variant call format (VCF) 20200815 CeNDR release. These variants were pruned to the 598,408 biallelic single nucleotide variants (SNVs) in non-divergent regions and without missing genotypes. Genetic distance among the 207 wild strains was calculated using the function *dist()* in R. Expression distance among the 207 wild strains was calculated based on the mean expression of 22,268 transcripts in non-divergernt regions and without missing data using the same *dist()* function in R. The unrooted neighbor-joining trees for genetic and expression relatedness were made using the R packages *phangorn* (v2.5.5)[82], *ape* (v5.6)[83] and *ggtree* (v1.14.6)[84].

**eQTL mapping**

*Input phenotype and genotype data.* For the 25,849 transcripts, we summarized the expression abundance of replicates to have the mean expression for each transcript of each strain as phenotypes used in GWA mapping (Supplementary Data 1). Genotype data for each of the 207 strains were acquired from the hard-filtered isotype VCF (20200815 CeNDR release).

*Permutation-based FDR threshold.* We performed GWA mapping using the pipeline *cegwas2-nf* (https://github.com/AndersenLab/cegwas2-nf). The pipeline uses the eigen-decomposition significance (EIGEN) threshold or the more stringent Bonferroni-corrected significance (BF) threshold to correct for multiple testing because of the large number of genetic markers (SNVs). To further correct for false positive QTL because of the large number of transcript expression traits, we computed a permutation-based False Discovery Rate (FDR) at 5%. We randomly selected 200 traits from our input phenotype file and permuted each of them 200 times. These 40,000 permuted phenotypes were used as input to call QTL using *cegwas2-nf* with EIGEN and BF threshold, respectively, as previously described[47,49,52]. Briefly, we used *BCFtools*[73] to filter variants that had any missing genotype calls and variants that were below the 5% minor allele frequency. Then, we used *-indep-pairwise 50 10 0.8* in *PLINK* v1.9[85,86] to prune the genotypes to 20,991 markers with a linkage disequilibrium (LD) threshold of $r^2 < 0.8$ and then generated the kinship matrix using the *A.mat()* function in the R package *rrBLUP* (v4.6.1)[87]. The number of independent tests ($N_{test}$) within the genotype matrix was estimated using the R package *RSpectra* (v0.16.0) (https://github.com/yixuan/RSpectra) and *correlateR* (0.1) (https://github.com/AEBilgrau/correlateR). The eigen-decomposition significance (EIGEN) threshold was calculated as $-\log_{10}(0.05/N_{test})$. We used the *GWAS()* function in the *rrBLUP* package to perform the genome-wide mapping with the EMMA algorithm[32]. QTL

 

were defined by at least one marker that was above the EIGEN or BF threshold. The EIGEN and BF 5% FDR were calculated as the 95 percentile of the significance of all the detected QTL under each threshold. The EIGEN and BF 5% FDR thresholds were 6.11 and 7.76, respectively.

**eQTL mapping.** We performed GWA mapping on the expression traits of the 25,849 transcripts as for permuted expression traits but using the EIGEN 5% FDR (6.11) as the threshold. We identified QTL with significance that also passed the Bonferroni 5% FDR threshold to locate the best estimate of QTL positions with the highest significance. We used the generic function *cor()* in R and Pearson correlation coefficient to calculate the phenotypic variance explained by each QTL. We used the *LD()* function from the R package *genetics* (v1.3.8.1.2) (https://cran.r-project.org/package=genetics) to calculate the LD correlation coefficient $r^2$ among QTL for traits with multiple eQTL.

**eQTL classification.** Local eQTL were classified if the QTL was within a 2 Mb region surrounding the transcript (+/−1 Mb from the transcript start site). All other QTL were classified as distant.

**Heritability calculation.** Heritability estimates were calculated for each of the 25,849 transcript expression traits used for eQTL mapping as previously described[88]. Narrow-sense heritability ($h^2$) was calculated with the phenotype file and pruned genotypes in eQTL mapping using the functions *mmer()* and *pin()* in the R package *sommer* (v4.1.2)[89]. Broad-sense heritability ($H^2$) was calculated using the expression of replicates of each strain and the *lmer* function in the R package *lme4* (v1.1.21) with the model phenotype ~1 + (1|strain)[90].

**Hotspot identification.** We first filtered out distant eQTL in common hyper-divergent genomic regions of wild *C. elegans* strains. Common hyper-divergent regions were defined among our 206 strains (reference N2 excluded) as described previously[27]. Briefly, we divided the genome into 1 kb bins and calculated the percentage of 206 strains that are classified as hyper-divergent in each bin. Common hyper-divergent regions were defined as bins ≥5%[27].

Distant eQTL hotspots were identified by dividing the genome into 0.5 cM bins and counting the number of non-divergent distant eQTL that mapped to each bin. Significance was determined as bins with more eQTL than the 99th percentile of a Poisson distribution using the maximum likelihood method and the function *eqpois()* in the R package *EnvStats* (v2.3.1)[1,3,9,91]. In total, 67 hotspots (0.5 cM each) were identified. We further merged hotspots that were located immediately next to each other to have 46 hotspots.

**Reanalysis of RIAILs eQTL.** We reclassified eQTL detected in a previous study using microarray expression data from synchronized young adult populations of 208 recombinant inbred advanced intercross lines (RIAILs) derived from N2 and CB4856[3,9,92]. A total of 2540 eQTL from 2196 probes were identified using linkage mappings[9]. We selected 2387 eQTL of 2054 probes that are in 2003 live genes based on the probe-gene list in the R package *linkagemapping* (v1.3) (https://github.com/AndersenLab/linkagemapping) and the GTF file (WS276)[53]. We classified 1208 local eQTL and 1179 distant eQTL as described above. We further identified and merged hotspots as above for 1124 distant eQTL that are not in the hyper-divergent regions of CB4856.

**Identification of overlapping eQTL among conditions/studies.** To identify overlapping eQTL in this study with previous studies[3,9,17–23], we obtained raw mapping data of each study from WormQTL2 (https://www.bioinformatics.nl/EleQTL/)[15], except the eQTL list of one recent study[22] from its online supplementary files and the RIAILs eQTL we reanalyzed above. For raw mapping data that include LOD scores of genome-wide markers for each gene, we used thresholds from the original papers to determine significant eQTL for each study. Then, we identified eQTL markers that overlapped with eQTL regions of interest from this study. For the eQTL list of the recent study[22], we identified eQTL intervals that overlapped with eQTL regions of interest from this study.

**Fine mapping for causal genes underlying hotspots.** For transcript expression traits with distant eQTL in hotspots, we performed fine mapping using the pipeline *cegwas2-nf* as previously described[47]. Briefly, we defined QTL regions of interest from the GWA mapping as +/−100 SNVs from the rightmost and leftmost markers above the EIGEN 5% FDR significance threshold. Then, using genotype data from the imputed hard-filtered isotype VCF (20200815 CeNDR release), we generated a QTL region of interest genotype matrix that was filtered as described above, with the one exception that we did not perform LD pruning. We used *PLINK* v1.9[85,86] to extract the LD between the markers used for fine mapping and the QTL peak marker identified from GWA mappings. We used the same command as above to perform fine mappings. To identify causal genes and variants that affect the expression of several transcripts underlying hotspots, we retained the fine-mapped candidate variants that passed the following per QTL per trait filters: top 5% most significant markers; out of common hyper-divergent genomic regions; with negative BLOSUM[93] scores as characterized and annotated in CeNDR[25].

**Enrichment analysis.** Gene set enrichment analyses were carried out for all genes found with transcript-level eQTL and for genes with distant eQTL in each hotspot using the web-based tool *WormCat*[38].

**Mediation analysis**

*GWA mapping of different* C. elegans *phenotypes.* We obtained nine different phenotype data used in previous *C.elegans* natural variation and GWA studies[13,42,46–52]. We filtered genetically distinct isotype strains for each trait based on CeNDR (20200815 release) and performed GWA mapping as for permuted expression traits but mostly using EIGEN or BF as the threshold according to the original studies. GWA was performed under EIGEN for two studies originally using BF as the threshold[48,49]. To summarize, EIGEN thresholds were used in GWA QTL identification for responses to arsenic[47] (Fig. 5b), zinc[13] (Fig. 5c), etoposide[48] (Fig. 5d), propionate[49] (Fig. 5e), abamectin[50] (Fig. 5f), and dauer formation in response to pheromone;[51] BF thresholds were used in GWA QTL identification for response to albendazole (Supplementary Fig. 6), telomere length[46] (Fig. 5a), and lifetime fecundity[52] (Fig. 5g).

*Mediation analysis.* For each QTL of the above phenotypes, we used the genotype (*Exposure*) at the phenotype QTL peak, transcript expression traits (*Mediator*) that have eQTL overlapped with the phenotype QTL, and the phenotype (*Outcome*) as input to perform mediation analysis using the *medTest()* function and 1000 permutations for *p*-value correction in the R package *MultiMed* (v2.6.0) (https://bioconductor.org/packages/release/bioc/html/MultiMed.html). For mediation, we used only strains with all of the three input data types available and where variation was found. For instance, between the 202 strains used in the study of ABZ resistance[42] and the 207 strains used in this study, 167 strains overlapped. Although we searched overlapped eQTL against QTL in the GWA mapping for ABZ resistance using 202 strains (Supplementary Fig. 6), 167 strains at maximum were used in mediation analysis. Furthermore, because some transcripts were found in hyper-divergent regions in certain strains and their expression data were filtered out, the rest of the strains with all of the data types available might contain no variation in one or all of the three data types and were not used in mediation analysis. For example, we found 1193 eQTL overlapped with the QTL on chromosome II in the GWA mapping for ABZ resistance, but only 1157 mediation analyses were performed.

The R package *MultiMed* could calculate the mediation effects of multiple mediators and the adjusted *p*-values efficiently, but it does not provide estimates of the "total effect" (the estimated effect of genotype on phenotype, ignoring expression) or the "proportion" of mediation effect in the total effect. This "proportion" should be non-negative and less than or equal to 1. We have previously[9] used the R package *mediation* (version 4.5.0)[94] to estimate "total effect" and "proportion". Mediators with negative proportion values or those higher than 1 were classified as uninterpretable and dropped. However, calculation using *mediation* is more time-consuming than using *MultiMed*. Therefore, we performed a second mediation analysis using the *mediate()* function in *mediation* only for significant mediators (adjusted *p* < 0.05 or mediation estimate greater than the 99th percentile of the distribution of mediation estimates) in the results of *MultiMed*. Then, we filtered out mediators with the uninterpretable results.

*GWA of traits regressed by transcript expression.* We regressed the trait animal length (q90.TOF)[42] by expression of every transcript using the generic function *residuals()* in R, which fits a linear model with the formula (*phenotype ~ expression*) to account for any differences in phenotype parameters present in transcript expression. Then GWA was performed for each regressed trait as for permuted expression traits using BF as the threshold.

**Reporting summary.** Further information on research design is available in the Nature Research Reporting Summary linked to this article.

## Data availability

The raw RNA-seq data generated in this study have been deposited in the NCBI Sequence Read Archive under accession code PRJNA669810. The raw expression counts and TPM quantified in this study have been deposited in the NCBI's Gene Expression Omnibus under accession code GSE186719. The datasets for generating all figures are available at https://github.com/AndersenLab/WI-Ce-eQTL. The *C. elegans* reference genome (WS276) and the GTF file (WS276) were obtained from WormBase (ftp://ftp.wormbase.org//pub/wormbase/releases/WS276/species/c_elegans/PRJNA13758). The hard-filtered isotype VCF (20200815 release) was obtained from CeNDR (https://www.elegansvariation.org/data/release/20200815). The transposable element sequences of *C. elegans* were obtained from Dfam (release 3.3) (https://www.dfam.org/releases/Dfam_3.3/). Previous *C. elegans* eQTL data were obtained from WormQTL2 (https://www.bioinformatics.nl/EleQTL/?mode=download) and the original papers.

## Code availability

The RNA-seq mapping pipeline can be found at https://github.com/AndersenLab/PEmRNA-seq-nf[95]. The mediation analysis pipeline can be found at https://github.com/

AndersenLab/mediation_GWAeQTL[96]. The code for generating all figures can be found at https://github.com/AndersenLab/WI-Ce-eQTL[97].

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

## Acknowledgements

We would like to thank Stefan Zdraljevic and Samuel J. Widmayer for helpful suggestions. G.Z. is supported by the NSF-Simons Center for Quantitative Biology at Northwestern University (awards Simons Foundation/SFARI 597491-RWC and the National Science Foundation 1764421). S.R.H. was funded by a DFG fellowship (HA 8449/1-1) from the Deutsche Forschungsgemeinschaft (www.dfg.de). E.C.A. is supported by a National Science Foundation CAREER Award (IOS-1751035) and a grant from the National Institutes of Health R01 DK115690. The *C. elegans* Natural Diversity Resource is supported by a National Science Foundation Living Collections Award to E.C.A. (1930382). We would also like to thank WormBase without which these analyses would not have been possible.

## Author contributions

E.C.A. conceived of and designed the study. D.L., S.R.H., and G.Z. prepared *C. elegans* cultures. G.Z. and N.M.R. performed RNA-seq experiments. G.Z. analyzed the data. G.Z. and E.C.A. wrote the manuscript.

## Competing interests

The authors declare no competing interests.
