## [Peer Review File · Nature Communications]

The impact of species-wide gene expression variation on
Caenorhabditis elegans complex traitsReviewers' Comments:

Reviewer #1:

Remarks to the Author:

The authors have provided an atlas of gene expression variation of wild strain *C. elegans*. The resource is valuable as this is the first study of natural variations of *C. elegans* transcriptome. The dataset can certainly help more researchers that work with *C. elegans* to understand the genetic variations of organismal level traits. With that said, there are major concerns about the data analysis:

1. In "Introduction", the authors reasoned only "a tiny fraction of the natural diversity of gene expression" can be revealed from genetic analysis of laboratory strains. The dataset actually provided a great opportunity to examine the hypothesis. Can the authors provide a comparison of the variations observed in wild strains and to that of laboratory strains from previous studies? For example, what is the estimated heritability of each transcript expression in wild strains and the laboratory crosses?
 2. To continue with my last point, a systematic comparison of the eQTLs, both cis- and distal, revealed from the current study and that of intercross lines (RIAILs) is necessary. Currently, the authors only showed the eQTLs distributed similarly in chromosome parts (arms versus centers), which is not insightful. It would be more interesting to show whether the eQTLs of the same transcript coincide between studies?
 3. The eQTL analysis is an important component of the work. However, the descriptions in Method is sometimes confusing. For example, in "selection of reliably expressed transcripts", the authors said 3,775 transcripts were "filtered out" because they are in hyper-divergent genomic regions, and then in the same paragraph, they said "further excluded 194 transcripts in hyper-divergent regions". What are the differences, and why are these 194 transcripts not filtered at the first place? I suggest the authors to provide a flowchart to describe the every step in QC of transcripts and samples, and clearly provide the number of transcripts/samples/strains removed and retained at each step.
 4. For the enrichment analysis, are the chromosome positions containing distal eQTLs enriched with transcription factors and chromatin factors?
 5. The mediation analysis of gene expression regarding to ABZ response is interesting. However, the analysis and results is valid only with the assumption that the transcriptome is unaffected with treatment of ABZ, which is unlikely the case. Analysis of organism level traits in normal laboratory conditions would not suffer from this problem. The authors should consider a better application for the mediation analysis.
 6. The interpretation of Figure 4C is problematic. It cannot exclude the possibility that the variant actually directly act in animal length, and then the animal length regulates the gene expression.
- Minor points:
7. How did the authors treat replicates of the same strain in the eQTL analysis? It is not clear in the manuscript.
 8. The authors used genes/transcripts/traits to describe expression traits. Sometimes it is not clear whether "traits" refer to gene level expression, or transcript level expression, or something else. It's better to use unified terms to avoid confusion.
 9. In discussions of estimated heritability (Page 6), the threshold 0.18 is mentioned several times, like the number of transcripts with h^2 larger than 0.18. It is not obvious why 0.18 is chosen.

Reviewer #2:

Remarks to the Author:

This manuscript introduces the compelling, largest resource to date of gene expression from naturally diverse *C. elegans* strains, employing sound methodology and analysis to describe the landscape of gene regulatory variation (eQTLs) in this species and to lay a foundation for improved quantitative genetics, especially mapping the causal genes and variants underlying diverse traits, in this keystone model organism. The authors extensively characterize the eQTL landscape; they detect eQTLs for 4250 genes (2655 local and 2382 distant), with up to 6 distant eQTL regulating a single gene; they

characterize the 67 detected distant eQTL hotspots, which comprise 54% of all distant eQTL and can regulate multiple classes of genes, including nominating candidate genes underlying distant eQTL hotspots and therefore regulating many genes; and they use their gene expression and eQTL datasets in mediation analysis to nominate causal genes for several organismal-level phenotypes by integrating the current dataset with previous GWA studies. The use of eQTLs for this mediation analysis is a primary motivator for the work.

This manuscript describes what will become a foundational resource to the large *C. elegans* research community, furthering the usefulness of this model organism in investigations of quantitative genetics and adding to our understanding of gene regulatory principles more broadly. The work described here will therefore make an already extraordinarily useful and productive model system more broadly useful, and accordingly is of great value and interest. Overall, I find the manuscript's conclusions important and well-supported, and their methods careful, sound, and appropriate. I do feel that many parts of the manuscript would substantially benefit from improved clarity of presentation prior to publication in a journal with a broad audience (see comments below).

Below, I lay out:

Major comments – things that warrant improved presentation and clarity, or occasionally analysis, prior to publication.

Minor comments – things that would improve the manuscript if addressed or would make the manuscript better conform to editorial requirements, but are not cause for concern

Proofreading corrections and stylistic suggestions – small observations of typos and the like and suggestions for stylistic improvements, included only in case helpful to the authors

....Major comments....

- A major strength of this work is its likely future use as a community resource. Are there plans to make these data easily query-able by the public, e.g., via CeNDR? If so, it would strengthen the manuscript to discuss these plans. (The data and code sharing already undertaken is commendable.) Indeed, more emphasis on the usefulness of this resource would improve the manuscript overall: the analyses are largely centered around mapping and quantitative resource building, rather than other expression data-enabled, biologically-driven analyses; therefore, leaning into and clarifying this focus would make for a stronger manuscript.

- The manuscript's title is about gene expression variation, but the manuscript's focus is not really on cataloging (in an atlas) the variation in gene expression, but rather more on mapping the genetic basis of gene expression variation (eQTLs) and using these data to map other complex traits. I recommend changing the title to reflect the main analyses and points of the paper, rather than reflecting something about the underlying dataset. (Only Fig 1e is really about variation in gene expression directly, rather than what underlies this genetically [Fig 2-3] and how it can benefit mapping [Fig 4-5])

- Developmental age (fig 1c). The synchronization and matching of stage via collection at first embryo is appropriate, making the range of ages determined with RaPTOR potentially surprising. This is not fully explained in the text (lines 103-110); the text seems to argue both that the age range observed is surprisingly wide and at the same time appropriately narrow, and furthermore this result's inclusion as part of the first main figure makes it seem important. I suggest re-organizing or re-describing this result.

- eQTLs, heritability, and genetic architecture of gene expression. The eQTL identification, counting, and classification part of this section (Fig 2b, d, e) is clear, but the points connecting eQTLs to heritability (Fig 2 a, c especially) do not come across as fully formed:

a) The methods used to estimate broad-sense and narrow-sense heritability don't necessarily capture H^2 and h^2 as understood by quantitative genetics purists, so the use of these terms (especially without additional contextualization) is potentially confusing/misleading.

b) There seems to be some circularity between defining narrow-sense heritability as what you can detect with these SNPs (if I'm understanding the methods correctly) and then concluding that eQTLs can be detected for traits with high 'heritability' at these SNPs.

The writeup surrounding line 158 wanders and seems conflicting, potentially due to point (b) above.

- Distant eQTL hotspot identification and characterization: Hotspots were identified as any 0.5cM bin of the genome with more eQTL than the Poisson-expected 99th percentile. This method seems great for initial bin identification, but I wonder if it would make sense to merge nearby bins, as it seems arbitrary to have several 'hotspots' in very close proximity to one another rather than one larger hotspot. I began thinking about this when reading the section about underlying causal genes and variants and becoming confused that the same gene would be causal across multiple hotspots before realizing that these 'multiple hotspots' were often adjacent to one another and/or close to one another. A merging of the initial arbitrarily binned hotspots and then GSEA and causal gene analysis within these hotspots would make the results more interpretable.

Relatedly, the analysis of distant eQTL hotspots' targets and gene content (paragraph starting line 230) is not entirely clear/convincing as written. Only 18 of the 67 hotspots are shown in the GSEA figure (Supp Fig. 5) – presumably because only these had significant enrichments, but this is not stated, nor is there discussion of whether this number (18 of 67) is meaningful or interesting. The results in this paragraph are also largely descriptive, leaving the reader wondering if any of this is surprising. Perhaps further interpretation or a de-emphasis of these results would help. Additionally, the overlap with known/predicted chromatin cofactors and TFs (same paragraph and Supplementary Fig. 6) is hard to interpret: is this significant overlap? Random? Etc.

- Mediation analysis. This is a major, probably the major, rationale of the paper, and so deserves and needs to be further explained. This section would benefit from a more thorough introduction to mediation analysis, as well as more explicit explanation of the results themselves, especially the ben-1 example. Some non-exhaustive specific questions and suggestions:

- Fig 4b (and related text). I am not (yet) convinced by the loss of correlation after regressing animal length on ben-1 expression: if any x and y are correlated, and then you look at residuals($\text{lm}(y \sim x)$) vs. x , there is no longer any correlation because you've regressed out their relationship. So, it does not seem particularly meaningful that ben-1 expression and length are negatively correlated, but ben-1 expression and residuals ($\text{lm}(\text{length} \sim \text{ben-1 expression})$) are not. Additionally here, the data points separate into two fairly distinct groups, those that have a ben-1 variant of interest and those that don't, and some of the correlation results seem to be driven by inter-group differences rather than the overall pattern. What does this mean for the interpretation of the figure? (In 4b right, within each group the correlation between ben-1 expression and animal length residuals after regression looks by eye to be positive.)

- The paragraph describing the mediation analyses on 8 traits (starting line 351) would benefit from an overarching numerical summary: how many new causal genes are nominated? Previously identified causal genes? Breakdown across traits? (Perhaps there's room for a table as part of figure 5?). Something to support or lead into a broader conclusion would be helpful.

- The removal of genes in hyperdivergent regions. This choice is well-defended and sensibly explained in the text but would benefit from more discussion of the types of result bias this may introduce. A brief paragraph somewhere speculating on what may or may not be systematically missed by excluding these genes (and SNVs for eQTL analysis) would be helpful, rather than just mentioning it's a constraint; the authors could draw on their earlier paper characterizing these haplotypes and their gene content (Lee et al 2021).

(minor related comment) Additionally, referring to this filtering where already particularly relevant would be helpful: For example – at line 122/Fig 1e (relative closeness of expression data vs. SNP data), the interpretation that "stabilizing selection has constrained variation in gene expression" follows logically from the figure in my opinion, but I wonder if the removal of the transcripts in hyperdivergent regions might be removing some of the most variable genes. The Methods make clear that both the SNP data (Fig 1d) and expression data (Fig 1e) both had hyperdivergent regions

removed; this seems important enough to the interpretation of the figure to make clear in the main text.

.... Minor comments....

- Line 122/Fig 1e: While there are many analyses that could be performed with this gene expression dataset, the current paper focuses on the utility of the dataset for mapping gene expression variation and other downstream traits. How does this (potentially interesting!) result about the relative closeness of expression data vs. SNP data and the interpretation of stabilizing selection relate to the broader focus of the paper? Why is this particular analysis included?

- line 179 and 690, local eQTLs definition is 'within a two mega base region'. Recommend making 100% clear that this means +/- 1Mb from the transcript.

- Line 55: "although a substantial amount of eQTL have been identified in different species, it is still largely unknown how gene expression variation relates to organism-level phenotypic differences." And then line 286 "...gene expression has been found to play an intermediate role between genotypes and phenotypes." These seem at least mildly contradictory, and neither is cited. In addition to reconciling these, I also recommend qualifying 'phenotypes' in line 287 with 'organismal' or similar as you did previously, given that gene expression is itself a phenotype.

- Supp Figs 2 and 5 (GSEA figures) - it is confusing that gene classes are represented more than once on the figures; an explanation in the legend would be helpful (I imagine this is a particularity of the tool used?). The duplication of gene classes across major categories also makes the results harder to evaluate.

- Supp Fig 3 and the independence of eQTLs: this is an important analysis, but it's hard to interpret without knowing what the background pairwise LD is. It would be helpful to add what pairwise LD looks like among randomly selected SNVs, and to show proportion rather than number of pairs on the Y axis so that the 'on the same chromosome' observations can be seen (or, split the histograms). In my opinion this point about independence vs. not of eQTLs deserves more explication. Additionally, 'LD values' language is used, recommend being explicit that these are R² values in the figure itself and the legend.

- Fig 4b (line 330) - add p-values for correlation coefficients

- Fig 5/line 378: It is a bit confusing that in the top plots, color and threshold are both related to significance, while in the bottom plots, the threshold is the 99th percentile but only the colored points are also statistically significant

- Supp Fig 4 and the discussion of this - It is not entirely clear what the reader is meant to take away from this figure; it might be helpful to more thoroughly explain (are the hotspots different from their local environment? What do you/would you expect to see if so?) or to remove the analysis. Additionally, it's somewhat confusing that there isn't higher nucleotide diversity evident on the arms when historical recombination and SNV burden is greater on the arms (as this manuscript demonstrates in Table 1)

- Supp Fig 7 - This fine mapping is impressive, but this is too much data for a reader to take in. Some related thoughts and questions in case helpful: Perhaps these could be displayed on a webpage where one locus at a time could be shown? Or narrow down the number of plots shown by some criteria?

Also, more information in the legend would be helpful, for example: Is each transcript shown only once per hotspot? How are transcripts grouped when the same hotspot is broken into two candidate genes? Could multiple candidate genes be highlighted together (i.e., in different colors) for all the transcripts in cases where there is multiple to help the reader build confidence about how the candidates were selected (to show that the transcripts for one candidate have different relationship to it than to another candidate)?

- line 585 (and anywhere else only number of transcripts is mentioned) - would be helpful to also report N genes, as is done throughout most of the paper

- Clarity in methods. Overall, the methods section and introduction to methods in results are reasonably clear, but two things in particular would benefit from expansion.

1) eQTL methods and results use two statistical significance thresholds without explaining why one isn't chosen and used consistently; the methods hint at this but a brief note would be helpful - at least to this reader - especially given this is pretty prominent: Fig 2c is split into two based on significance

threshold; similarly, Fig 5 has some panels with hits detected using one threshold and other panels using another, without any clear explanation this reader could easily find, and is using 2 thresholds for both GWA mapping and eQTL display.

2) Mediation analysis methods especially line 780 – a fuller explanation of the nested mediation analysis would be helpful: why did the first analysis give uninterpretable results that then needed to be re-estimated? If this is sensible mathematically as I imagine it is, it would be helpful to briefly explain why.

- Reporting summary – sample size. Here the number of animals per plate is discussed, but I believe it would make more sense to discuss how the number of strains sequenced was chosen – these are the distinct samples.

- Reporting summary/methods – it is not clear how strains were divided into batches or randomized, other than for RNA library construction where this is more thoroughly described in Methods. This should be clear at least in the reporting summary (were replicates from the same strain done on different days/different batches? How many total batches? etc)

....Proofreading corrections and stylistic suggestions....

- Fig 2: consider including a legend describing colors in the figure itself especially for (a) which has different colors than the rest of the figure

- Paragraph beginning line 476 is somewhat repetitive and has a different tone than the surrounding paragraphs; likely dispensable, or you could especially get rid of the last sentence.

- Recommend finding all the instances the adjective 'diverse' is used and replacing with synonyms at least some of the time, focusing on whether what is described can actually be 'diverse'

- line 675 typo, %5 should be 5%

- Line 806 acknowledgments, a couple missing/misplaced words, this is probably meant to read 'We would also like to thank WormBase without which these analyses would not have been possible.'

REVIEWER COMMENTS

Reviewer #1 (Remarks to the Author):

The authors have provided an atlas of gene expression variation of wild strain *C. elegans*. The resource is valuable as this is the first study of natural variations of *C. elegans* transcriptome. The dataset can certainly help more researchers that work with *C. elegans* to understand the genetic variations of organismal level traits. With that said, there are major concerns about the data analysis:

1. In "Introduction", the authors reasoned only "a tiny fraction of the natural diversity of gene expression" can be revealed from genetic analysis of laboratory strains. The dataset actually provided a great opportunity to examine the hypothesis. Can the authors provide a comparison of the variations observed in wild strains and to that of laboratory strains from previous studies? For example, what is the estimated heritability of each transcript expression in wild strains and the laboratory crosses?

We agree that this point is important, which is why we addressed it in Discussion. We compared our study with the eQTL study using N2xCB4856 recombinant inbred lines^{1,2}. Among the 6,545 eQTL that we detected, the strains N2 and CB4856 shared the same genotypes in 4,476 eQTL, which could not be discovered using N2xCB4856 recombinant inbred line linkage studies. On the other hand, the N2xCB4856 RIALs might provide greater power than our study, because 1,870 eQTL of 1,579 genes were only detected using the N2xCB4856 RIALs expression data. To expand that analysis, we compared our eQTL study using GWAS to all other known eQTL studies using linkage and NIL mappings along with environmental perturbations of wild strains (new Supplemental Figure 10). We also added a section to the Discussion to enrich this comparison. Please see the explanations below in response to point 2 and the Methods major concern. Heritability estimates are not comparable between different strain sets because they have differences in demography and allele frequencies, so we have not included those comparisons.

2. To continue with my last point, a systematic comparison of the eQTIs, both cis- and distal, revealed from the current study and that of intercross lines (RIALs) is necessary. Currently, the authors only showed the eQTLs distributed similarly in chromosome parts (arms versus centers), which is not insightful. It would be more interesting to show whether the eQTLs of the same transcript coincide between studies?

Again, we have also addressed this point in Discussion. Our study and the N2xCB4856 RIALs eQTL study detected overlapping local eQTL for 454 genes and distant eQTL for 19 genes. To further address this issue, we gathered eQTL data in *C. elegans* from another seven studies³⁻⁹. The seven studies, the N2xCB4856 RIALs eQTL study we mentioned above^{1,2}, and our study represent *C. elegans* eQTL studies using N2xCB4856 RILs, N2xCB4856 RIALs, JU1511xJU1926xJU1931xJU1941 multiparental RILs, and wild strains under 14 different conditions, such as different environmental conditions and different developmental stages¹⁰. We examined whether the eQTL we detected in our study had been identified in any of the eight studies and the total number of detections across all nine studies. These new data are in Supplemental Figure 10 and a new section in the Methods and Discussion.

"Identification of overlapping eQTL among conditions/studies"

To identify overlapping eQTL in this study with previous studies¹⁻⁹, we obtained raw mapping data of each study from WormQTL2 (<https://www.bioinformatics.nl/EleQTL/>)¹⁰, except the eQTL list of one recent study⁹ from its online supplementary files and the RIALs eQTL we reanalyzed above. For raw mapping data that include LOD scores of genome-wide markers for each gene, we used thresholds used in original papers to determine significant eQTL for each study. Then, we identified eQTL markers that overlapped with eQTL regions of interest from this study. For the eQTL list of the recent study⁹, we identified eQTL intervals that overlapped with eQTL regions of interest from this study."

Supp Fig. 8:

Supplementary Fig. 8

The genomic locations of 2,029 eQTL peaks (x-axis) that overlapped with eQTL detected in eight previous studies¹⁻⁹ are plotted against the genomic locations of the 1,993 transcripts of 1,625 genes with expression differences (y-axis). Golden points on the diagonal of the map represent local eQTL that colocalize with the transcripts that they influence. Purple points correspond to distant eQTL that are located further away from the transcripts that they influence. The size of each point represents the total number of detections for each eQTL in the 14 conditions of the nine studies, including this study.

“To further examine whether the eQTL we found here had been identified previously, we collected eQTL data in *C. elegans* from another seven studies³⁻⁹. These seven studies measured expression of N2xCB4856 RILs and JU1511xJU1926xJU1931xJU1941 multiparental RILs under different conditions and identified eQTL³⁻⁹. Our study, the above N2xCB4856 RILs eQTL study, and the seven studies represent expression variation under 14 different conditions, such as different environmental conditions and different developmental stages. We examined whether the eQTL we detected in our study had been identified in any of the eight studies and the total number of detections across all nine studies. We found that 2,029 eQTL for 1,993 transcripts of 1,625 genes found in our study were previously detected (Supplementary Fig. 8). The median number of detections of these 2,029 eQTL across the 14 conditions of nine studies is three (ranging from two to 14) (Supplementary Fig. 8). The overlapped eQTL identified across studies suggested small effects of developmental stages and environmental conditions on regulation of these genes.”

3. The eQTL analysis is an important component of the work. However, the descriptions in Method is sometimes confusing. For example, in "selection of reliably expressed transcripts", the authors said 3,775 transcripts were "filtered out" because they are in hyper-divergent genomic regions, and then in the same paragraph, they said "further excluded 194 transcripts in hyper-divergent regions". What are the differences, and why are these 194 transcripts not filtered at the first place? I suggest the authors to provide a flowchart to describe the every step in QC of transcripts and samples, and clearly provide the number of transcripts/samples/strains removed and retained at each step.

Hyper-divergent regions varied across *C. elegans* strains. We removed data of transcripts that were in the hyper-divergent regions on a per strain basis. The data of 3,775 transcripts were removed in at least one strain. After this filtering, the total number of strains that retained data in the 3,775 transcripts was lower than 207. To have relatively high power in eQTL mappings, we required that at least 100 strains should have retained data in each transcript. So we thoroughly filtered out 194 of the 3,775 transcripts that did not meet this final requirement.

We added the above statements to the Methods section to clarify these points. As suggested, we also indicated the numbers of transcripts, genes, samples, and strains from each step to the next step in the Supplementary Fig. 1.

4. For the enrichment analysis, are the chromosome positions containing distal eQTLs enriched with transcription factors and chromatin factors?

In the revision, we merged hotspots that were located immediately next to each other. The 67 hotspots were merged into 46 hotspots, with up to five hotspots merged into a single hotspot. We re-did the GSEA and causal gene analysis with these 46 hotspots. We used Fisher Exact tests to examine whether any of the 46 hotspots was enriched for TFs or chromatin cofactors. We found the hotspot at 30.5-33 cM on chromosome IV was enriched with chromatin cofactors (Fisher Exact Test, Bonferroni FDR corrected $p = 6 \times 10^{-5}$). A total of 34 genes encoding chromatin cofactors were found in this hotspot.

"Distant eQTL were not uniformly distributed across the genome. Of the 3,360 distant eQTL, 1,828 were clustered into 4667 hotspots, each of which affected the expression of 12 to 184 transcripts (Fig. 3). GSEA on genes with transcript-level distant eQTL in each hotspot revealed potential shared transcriptional regulatory mechanisms across different genes of the same class in 12 hotspots (Supplementary Fig. 4, Supplementary Data 3). We further examined the enrichment of genes encoding chromatin cofactors and transcription factors¹¹⁻¹³ in the region of each hotspot and found the hotspot at 30.5-33 cM on chromosome IV was enriched with chromatin cofactor genes (Fisher Exact Test, Bonferroni corrected $p = 6 \times 10^{-5}$). To identify any of these chromatin cofactor genes might be causal, we performed fine mapping on the 110 distant eQTL in this hotspot. We found that a linker histone chromatin cofactor gene, *hil-2*¹³, might underlie 33 of the 110 transcripts with distant eQTL in this hotspot. We further performed GSEA for these 33 transcripts and found enrichment in E3 ligases containing an F-box domain (Fisher Exact Test, Bonferroni FDR corrected $p = 0.003$) (Supplementary Fig. 5a), heat stress related genes (Fisher Exact Test, Bonferroni FDR corrected $p = 0.01$) (Supplementary Fig. 5b), and transcription factors of the homeodomain class (Fisher Exact Test, Bonferroni FDR corrected $p = 0.003$) (Supplementary Fig. 5c)."

5. The mediation analysis of gene expression regarding to ABZ response is interesting. However, the analysis and results is valid only with the assumption that the transcriptome is unaffected with treatment of ABZ, which is unlikely the case. Analysis of organism level traits in normal laboratory conditions would not suffer from this problem. The authors should consider a better application for the mediation analysis.

Yes, we agree that the transcriptome is likely to be affected by ABZ treatment, but statistical mediation analysis using expression data collected in normal conditions provides evidence on the baseline expression of *ben-1* and other genes that ultimately affect the response of animals to ABZ. The power of this approach is that it does not require the induction of expression differences to cause phenotypic differences (e.g., mediation

reflects potential functional differences in levels of genes that vary across strains). Future transcriptome data collected from animals with ABZ treatment could help us understand how gene expression changes in response to ABZ, but it is a different type of analysis. We clarified this point in the Discussion.

“Although the transcriptome is likely to be affected by ABZ treatment, statistical mediation analysis using expression data collected in normal conditions provided evidence on the baseline expression of *ben-1* and other genes that ultimately affected the response of animals to ABZ. The power of this approach is that it does not require the induction of expression differences to cause phenotypic differences (e.g., mediation reflects potential functional differences in levels of genes that vary across strains). Future transcriptome data collected from animals with ABZ treatment could help us understand how gene expression changes in response to ABZ.”

6. The interpretation of Figure 4C is problematic. It cannot exclude the possibility that the variant actually directly act in animal length, and then the animal length regulates the gene expression.

We have validated the causality of *ben-1* in *C. elegans* response to ABZ using the CRISPR-Cas9 genome editing system for multiple loci in *ben-1*¹⁴⁻¹⁶. The genetic variants in *ben-1* disrupt the function of BEN-1 to affect *C. elegans* responses to ABZ as shown by developmental stage (animal length)¹⁴⁻¹⁶. In this study, we found that the extreme allelic heterogeneity at *ben-1* might also underlie expression variation in *ben-1*. Because of the negative correlation between *ben-1* expression and animal length across strains, genetic variants in *ben-1* could affect the abundances of *ben-1* mRNA and BEN-1 protein to affect ABZ response in *C. elegans*. For these reasons, the changes in developmental stage (animal length) are likely to be the outcome instead of the causes of differential expression of *ben-1* in response to ABZ.

We added the following sentences prior to the interpretation of Fig. 4C in Results to support the conclusion.

“We have validated the causality of *ben-1* in *C. elegans* response to ABZ using the CRISPR-Cas9 genome editing system for multiple variants in *ben-1*¹⁴⁻¹⁶. These variants disrupt the function of BEN-1 to affect *C. elegans* responses to ABZ as shown by developmental stage (animal length) trait¹⁴⁻¹⁶.”

Minor points:

7. How did the authors treat replicates of the same strain in the eQTL analysis? It is not clear in the manuscript.

We described this point in the Methods, Line 649: “we summarized the expression abundance of replicates to have the mean expression for each transcript of each strain as phenotypes used in GWA mapping”.

8. The authors used genes/transcripts/traits to describe expression traits. Sometimes it is not clear whether "traits" refer to gene level expression, or transcript level expression, or something else. It's better to use unified terms to avoid confusion.

We went through all the occasions of “traits” and modified “traits” to “transcript expression traits” on Lines 148 and 694.

9. In discussions of estimated heritability (Page 6), the threshold 0.18 is mentioned several times, like the number of transcripts with h² larger than 0.18. It is not obvious why 0.18 is chosen.

The narrow-sense heritability threshold (0.18) was arbitrarily decided based on many previous studies in the Andersen lab. Overall, it is empirical and not justified theoretically. For this reason, we modified the sentence when it was first mentioned.

“However, h^2 of thousands of transcript expression traits indicated a substantial heritable genetic component of the population-wide expression differences.”

We also deleted the second sentence with “0.18” because the meaning and results are still clear without that arbitrary threshold.

Reviewer #2 (Remarks to the Author):

This manuscript introduces the compelling, largest resource to date of gene expression from naturally diverse *C. elegans* strains, employing sound methodology and analysis to describe the landscape of gene regulatory variation (eQTLs) in this species and to lay a foundation for improved quantitative genetics, especially mapping the causal genes and variants underlying diverse traits, in this keystone model organism. The authors extensively characterize the eQTL landscape; they detect eQTLs for 4250 genes (2655 local and 2382 distant), with up to 6 distant eQTL regulating a single gene; they characterize the 67 detected distant eQTL hotspots, which comprise 54% of all distant eQTL and can regulate multiple classes of genes, including nominating candidate genes underlying distant eQTL hotspots and therefore regulating many genes; and they use their gene expression and eQTL datasets in mediation analysis to nominate causal genes for several organismal-level phenotypes by integrating the current dataset with previous GWA studies. The use of eQTLs for this mediation analysis is a primary motivator for the work.

This manuscript describes what will become a foundational resource to the large *C. elegans* research community, furthering the usefulness of this model organism in investigations of quantitative genetics and adding to our understanding of gene regulatory principles more broadly. The work described here will therefore make an already extraordinarily useful and productive model system more broadly useful, and accordingly is of great value and interest. Overall, I find the manuscript’s conclusions important and well-supported, and their methods careful, sound, and appropriate. I do feel that many parts of the manuscript would substantially benefit from improved clarity of presentation prior to publication in a journal with a broad audience (see comments below).

Thank you for your critical comments and corrections! The manuscript is much improved from these suggestions.

Below, I lay out:

Major comments – things that warrant improved presentation and clarity, or occasionally analysis, prior to publication.

Minor comments – things that would improve the manuscript if addressed or would make the manuscript better conform to editorial requirements, but are not cause for concern

Proofreading corrections and stylistic suggestions – small observations of typos and the like and suggestions for stylistic improvements, included only in case helpful to the authors

....Major comments....

- A major strength of this work is its likely future use as a community resource. Are there plans to make these data easily query-able by the public, e.g., via CeNDR? If so, it would strengthen the manuscript to discuss these plans. (The data and code sharing already undertaken is commendable.) Indeed, more emphasis on the usefulness of this resource would improve the manuscript overall: the analyses are largely centered around mapping and quantitative resource building, rather than other expression data-enabled, biologically-driven analyses; therefore, leaning into and clarifying this focus would make for a stronger manuscript.

We completely agree with this point! Our plans are to build an eQTL browser and data query/download system into CeNDR, but this contract has not been completed yet. To advertise the future use of these data, we added this paragraph (below) to the end of Discussion.

“In addition to disseminating wild strains, CeNDR has also incorporated whole-genome alignment, annotated variant data, and various population genetic and genomic estimates in wild *C. elegans* strains^{17,18}. These data and the web-based tools, such as heritability estimation and GWA mappings, have facilitated many studies for the community. In the latest version of CeNDR, we have implemented the mediation analysis using the newly generated expression and eQTL data into the GWA mapping tool¹⁹ and provide the results along with other mapping outputs. We will further develop an expression browser on CeNDR for easy querying and interactive visualization of expression variation across wild strains. For genes and transcripts with detected eQTL, we will display the Manhattan plots and genotype-by-phenotype plots. We will also create tools to aid differential expression analysis and visualization between any pair of strains available in our data. We believe our data will further facilitate natural variation and evolutionary research in *C. elegans*.”

- The manuscript's title is about gene expression variation, but the manuscript's focus is not really on cataloging (in an atlas) the variation in gene expression, but rather more on mapping the genetic basis of gene expression variation (eQTLs) and using these data to map other complex traits. I recommend changing the title to reflect the main analyses and points of the paper, rather than reflecting something about the underlying dataset. (Only Fig 1e is really about variation in gene expression directly, rather than what underlies this genetically [Fig 2-3] and how it can benefit mapping [Fig 4-5])

We appreciate the reviewers concerns and changed the title of the paper to: The impact of gene expression variation on complex traits across the *Caenorhabditis elegans* species

- Developmental age (fig 1c). The synchronization and matching of stage via collection at first embryo is appropriate, making the range of ages determined with RaPTOR potentially surprising. This is not fully explained in the text (lines 103-110); the text seems to argue both that the age range observed is surprisingly wide and at the same time appropriately narrow, and furthermore this result's inclusion as part of the first main figure makes it seem important. I suggest re-organizing or re-describing this result.

Thank you for this suggestion. We re-organized and re-described this result.

“We harvested animals at the first embryo-laying event rather than at a certain age (hours post-hatching), because we observed variation in ages at the first embryo-laying event across strains. Additionally, we reasoned that expression is influenced primarily by the developmental stage. Here, we evaluated the age of each sample when they were harvested using our expression data and published time-series expression data²⁰. We inferred that our animals fit an expected developmental age of 60 to 72 hours post hatching (Fig. 1c), during which time the animal is in the young adult stage. The age estimation reflects natural variation in the duration from hatching to the beginning of embryo-laying of wild *C. elegans* strains.”

- eQTLs, heritability, and genetic architecture of gene expression. The eQTL identification, counting, and classification part of this section (Fig 2b, d, e) is clear, but the points connecting eQTLs to heritability (Fig 2 a, c especially) do not come across as fully formed:

a) The methods used to estimate broad-sense and narrow-sense heritability don't necessarily capture H2 and h2 as understood by quantitative genetics purists, so the use of these terms (especially without additional contextualization) is potentially confusing/misleading.

We agree that it is confusing to the purists, but these values are what the *C. elegans* field has established and used for many years. We added more information about strain replication and how we use the genetic relatedness matrix to help clarify these points. However, we are open to further suggestions!

b) There seems to be some circularity between defining narrow-sense heritability as what you can detect with these SNPs (if I'm understanding the methods correctly) and then concluding that eQTLs can be detected for traits with high 'heritability' at these SNPs.

We deleted the sentence on line 148 to avoid the circularity.

The writeup surrounding line 158 wanders and seems conflicting, potentially due to point (b) above.

We deleted the sentence from line 159 to line 161.

- Distant eQTL hotspot identification and characterization: Hotspots were identified as any 0.5cM bin of the genome with more eQTL than the Poisson-expected 99th percentile. This method seems great for initial bin identification, but I wonder if it would make sense to merge nearby bins, as it seems arbitrary to have several ‘hotspots’ in very close proximity to one another rather than one larger hotspot. I began thinking about this when reading the section about underlying causal genes and variants and becoming confused that the same gene would be causal across multiple hotspots before realizing that these ‘multiple hotspots’ were often adjacent to one another and/or close to one another. A merging of the initial arbitrarily binned hotspots and then GSEA and causal gene analysis within these hotspots would make the results more interpretable.

We merged hotspots that were located immediately next to each other following previous studies^{1,21}. Using this new approach, the 67 hotspots were merged into 46 hotspots, with up to five hotspots merged into a single hotspot. We re-did the GSEA and causal gene analysis with these 46 hotspots.

We also merged hotspots for eQTL identified in the linkage mapping studies (Supplementary Fig. 7b).

As suggested at the next point, we de-emphasized the enrichment results, removed Supplementary Fig. 6, and re-wrote the three paragraphs in the hotspot section.

“Distant eQTL were not uniformly distributed across the genome. Of the 3,360 distant eQTL, 1,828 were clustered into 46 hotspots, each of which affected the expression of 12 to 184 transcripts (Fig. 3). GSEA on genes with transcript-level distant eQTL in each hotspot revealed potential shared transcriptional regulatory mechanisms across different genes of the same class in 12 hotspots (Supplementary Fig. 4, Supplementary Data 3). We further examined the enrichment of genes encoding chromatin cofactors and transcription factors^{11–13} in the region of each hotspot and found the hotspot at 30.5-33 cM on chromosome IV was enriched with chromatin cofactor genes (Fisher Exact Test, Bonferroni corrected $p = 6\text{-E}5$). To suggest if any of these chromatin cofactor genes might be causal, we performed fine mapping on the 110 distant eQTL in this hotspot. We found that a linker histone chromatin cofactor gene, *hil-2*¹³, might underlie 33 of the 110 transcripts with distant eQTL in this hotspot. We further performed GSEA for these 33 transcripts and found enrichment in E3 ligases containing an F-box domain (Fisher Exact Test, Bonferroni FDR corrected $p = 0.003$) (Supplementary Fig. 5a), heat stress related genes (Fisher Exact Test, Bonferroni FDR corrected $p = 0.01$) (Supplementary Fig. 5b), and transcription factors of the homeodomain class (Fisher Exact Test, Bonferroni FDR corrected $p = 0.003$) (Supplementary Fig. 5c). Additionally, we performed fine mapping on distant eQTL in all the other hotspots and filtered for the most likely candidate variants (see Methods for details) (Supplementary Data 4). Then, we focused on the filtered candidate variants that were mapped for at least four transcript expression traits in each hotspot and are in genes encoding transcription factors or chromatin cofactors. In total, we identified 50 candidate genes encoding transcription factors or chromatin cofactors for 25 hotspots. For example, the gene *ttx-1*, which encodes a transcription factor necessary for thermosensation in the AFD neurons^{22,23}, might underlie the expression variation of 97 transcripts with distant eQTL in a 1.5 cM hotspot between (44.5-46 cM) on chromosome V. TTX-1 regulates expression of *gcy-8* and *gcy-18* in AFD neurons^{22,23}, but no eQTL were detected for the two genes likely because we measured the expression of whole animals. Besides the 50 candidate genes, the hundreds of other fine mapping candidates are not as transcription factors or chromatin cofactors, suggesting other mechanisms underlying distant eQTL. Altogether, as previously implicated in other species^{24–26}, our results indicate that a diverse collection of molecular mechanisms likely cause gene expression variation in *C. elegans*.”

Relatedly, the analysis of distant eQTL hotspots’ targets and gene content (paragraph starting line 230) is not entirely clear/convincing as written. Only 18 of the 67 hotspots are shown in the GSEA figure (Supp Fig. 5) – presumably because only these had significant enrichments, but this is not stated, nor

is there discussion of whether this number (18 of 67) is meaningful or interesting. The results in this paragraph are also largely descriptive, leaving the reader wondering if any of this is surprising. Perhaps further interpretation or a de-emphasis of these results would help. Additionally, the overlap with known/predicted chromatin cofactors and TFs (same paragraph and Supplementary Fig. 6) is hard to interpret: is this significant overlap? Random? Etc.

We clarified that “Only hotspots with significant enrichments were shown.” in the legend of Supp Fig. 5.

- Mediation analysis. This is a major, probably the major, rationale of the paper, and so deserves and needs to be further explained. This section would benefit from a more thorough introduction to mediation analysis, as well as more explicit explanation of the results themselves, especially the *ben-1* example. Some non-exhaustive specific questions and suggestions:

We expanded and re-wrote the introduction to mediation analysis and the description for results of *ben-1* and Fig. 4a:

“Mediation analysis seeks to identify the mechanism that underlies the relationship between an exposure (an independent variable) and an outcome (a dependent variable) via the inclusion of one or multiple mediators (intermediary mediating variables). The total effects of the exposure on the outcome include both direct effects that could not be explained by mediators and indirect effects that act through mediators. In quantitative genetics mapping studies, genotypes could affect organism-level phenotypes directly or indirectly through the intermediate effects of gene expression²⁵. Therefore, we could use mediation analysis to understand how genetic variants (exposure) affect organism-level phenotypic variation (outcome) through expression variation of one or multiple genes (mediators).”

“We performed mediation analysis on the animal length variation (outcome) in response to ABZ, the genetic variation (exposure) at the GWA QTL of the animal length variation, and the expression of 1,157 transcripts (potential mediators) that had eQTL that overlapped with the QTL for the animal length variation. We identified significant mediation effects by the expression of 12 transcripts of 11 genes, including *ben-1* (Fig. 4a). The expression of *ben-1* showed the second highest mediation effect among all the 12 mediators and explained 26% of the total effects via genetic variation on animal length variation.”

- Fig 4b (and related text). I am not (yet) convinced by the loss of correlation after regressing animal length on *ben-1* expression: if any x and y are correlated, and then you look at residuals($\text{Im}(y-x)$) vs. x , there is no longer any correlation because you’ve regressed out their relationship. So, it does not seem particularly meaningful that *ben-1* expression and length are negatively correlated, but *ben-1* expression and residuals ($\text{Im}(\text{length} \sim \text{ben-1 expression})$) are not. Additionally here, the data points separate into two fairly distinct groups, those that have a *ben-1* variant of interest and those that don’t, and some of the correlation results seem to be driven by inter-group differences rather than the overall pattern. What does this mean for the interpretation of the figure? (In 4b right, within each group the correlation between *ben-1* expression and animal length residuals after regression looks by eye to be positive.)

We agree the regression was not very meaningful. So we turned to regress *ben-1* expression by the existence of genetic variants in *ben-1* and found no correlation between regressed expression and animal length, further supporting that the allelic heterogeneity altered *ben-1* expression to affect ABZ response in *C. elegans*.

“We regressed *ben-1* expression by the existence of these genetic variants in *ben-1* and found no correlation between regressed expression and animal length (Fig. 4b), further supporting that the allelic heterogeneity altered *ben-1* expression to affect ABZ response in *C. elegans*.”

- The paragraph describing the mediation analyses on 8 traits (starting line 351) would benefit from an overarching numerical summary: how many new causal genes are nominated? Previously identified

causal genes? Breakdown across traits? (Perhaps there's room for a table as part of figure 5?). Something to support or lead into a broader conclusion would be helpful.

Causal genes that partially explained the phenotypic variation in all the traits, except for lifetime fecundity, have been identified and validated previously (Line 355, 356). The newly nominated candidates by mediation analysis were all indicated in Fig. 4a and Fig. 5. There are only few mediator genes for most of the traits, except 11 for ABZ response, 11 for abamectin response, and 17 for lifetime fecundity.

To compare with candidates from mediation analysis and fine mappings for lifetime fecundity, we added these analyses and descriptions:

“To compare with mediation results for lifetime fecundity, we performed fine mapping and identified top candidate genes as described for distant eQTL. We identified 74 candidate genes using fine mapping, without overlapped genes with the 17 mediator genes. Among these 17 mediator genes, seven genes, including *ets-4*, are on different chromosomes from the related QTL, suggesting that mediation analysis nominated new candidate genes that were unable to be detected in fine mappings.”

- The removal of genes in hyperdivergent regions. This choice is well-defended and sensibly explained in the text but would benefit from more discussion of the types of result bias this may introduce. A brief paragraph somewhere speculating on what may or may not be systematically missed by excluding these genes (and SNVs for eQTL analysis) would be helpful, rather than just mentioning it's a constraint; the authors could draw on their earlier paper characterizing these haplotypes and their gene content (Lee et al 2021).

Thank you for these suggestions. We added these points in Discussion.

“Genes in hyper-divergent regions were enriched in classes that were related to sensory perception, immune response, and xenobiotic stress response²⁷. Our data might not capture the full landscape of expression variation in these genes, potentially including some most variable genes, and their local regulatory loci.”

“Hyper-divergent regions might harbor the least proportion of well characterized SNVs in the genome. Therefore, the number of regulatory loci in hyper-divergent regions might be underestimated for both local and distant eQTL. Future efforts using long-read sequencing are necessary to study the sequence, expression, natural selection, and evolution of genes in hyper-divergent regions, which could improve the understanding on adaptation of *C. elegans* in various environments.”

(minor related comment) Additionally, referring to this filtering where already particularly relevant would be helpful: For example – at line 122/Fig 1e (relative closeness of expression data vs. SNP data), the interpretation that “stabilizing selection has constrained variation in gene expression” follows logically from the figure in my opinion, but I wonder if the removal of the transcripts in hyperdivergent regions might be removing some of the most variable genes. The Methods make clear that both the SNP data (Fig 1d) and expression data (Fig 1e) both had hyperdivergent regions removed; this seems important enough to the interpretation of the figure to make clear in the main text.

The original Fig 1d was not built using SNV data with hyper-divergent regions removed. However, we took the suggestion and filtered the 851,105 SNVs to 598,408 SNVs. We then built the tree the exact same way as for expression by calculating distance first. The new SNV tree is less divergent than the original one, but still shows geographical clustering and those extremely divergent strains.

We stressed that both SNVs and transcripts used for relatedness analyses were in non-divergent regions in the legend of Fig. 1.

.... **Minor comments....**

- Line 122/Fig 1e: While there are many analyses that could be performed with this gene expression dataset, the current paper focuses on the utility of the dataset for mapping gene expression variation and other downstream traits. How does this (potentially interesting!) result about the relative closeness of expression data vs. SNP data and the interpretation of stabilizing selection relate to the broader focus of the paper? Why is this particular analysis included?

The tree (Fig 1d) using SNPs was plotted to show genetic diversity of the 207 strains in the study. Because we are always interested in the population structure and genetic diversity in *C. elegans*, we compared the SNP tree and the expression tree (Fig 1e), which also served as an overview of expression variation across *C. elegans* wild strains. Furthermore, compared to variation in SNPs, variation in expression across strains might better reflect variation in organism-level phenotypes. Altogether, we applied this analysis.

- line 179 and 690, local eQTLs definition is 'within a two mega base region'. Recommend making 100% clear that this means +/- 1Mb from the transcript.

We modified this point as suggested.

- Line 55: "although a substantial amount of eQTL have been identified in different species, it is still largely unknown how gene expression variation relates to organism-level phenotypic differences." And then line 286 "...gene expression has been found to play an intermediate role between genotypes and phenotypes." These seem at least mildly contradictory, and neither is cited. In addition to reconciling these, I also recommend qualifying 'phenotypes' in line 287 with 'organismal' or similar as you did previously, given that gene expression is itself a phenotype.

We modified the two sentences and 'phenotypes' as suggested.

"Although a substantial amount of eQTL have been identified in different species, only few studies have addressed how gene expression variation related to organism-level phenotypic differences^{2,28-30}."

"In quantitative genetics mapping studies, genotypes could affect organism-level phenotypes directly or indirectly through the intermediate effects of gene expression²⁵."

- Supp Figs 2 and 5 (GSEA figures) - it is confusing that gene classes are represented more than once on the figures; an explanation in the legend would be helpful (I imagine this is a particularity of the tool used?). The duplication of gene classes across major categories also makes the results harder to evaluate.

We added the following sentences in the legend of Supp Fig 2:

"Wormcat¹³ provides enrichment results in broad categories (those in Category 1) and more specific categories (in Category 2 and 3). For example, the top enrichment in Category 1 is the proteolysis proteasome, more specifically, the ubiquitin ligases E3 of proteolysis proteasome in Category 2 and the F-box protein of ubiquitin ligases E3 of proteolysis proteasome in Category 3."

- Supp Fig 3 and the independence of eQTLs: this is an important analysis, but it's hard to interpret without knowing what the background pairwise LD is. It would be helpful to add what pairwise LD looks like among randomly selected SNVs, and to show proportion rather than number of pairs on the Y axis so that the 'on the same chromosome' observations can be seen (or, split the histograms). In my opinion this point about independence vs. not of eQTLs deserves more explication.

The background pairwise LD in *C. elegans* has been calculated previously (Andersen et al. 2012). As briefly mentioned in the manuscript, the previous study found strong LD ($r^2 > 0.6$) in several genomic regions on the same chromosomes and substantial LD ($r^2 > 0.2$) between chromosomes in wild *C. elegans* strains. Compared to these results and our experience in GWA mappings, we concluded that most eQTL underlying

expression variation of the same transcripts were likely independent. If we just randomly select SNVs across the genome, most pairwise LDs among the SNVs should be very low. Below is a figure showing pairwise LD (r^2) among 500 randomly selected SNVs across the genome. Most of the LD (r^2) values are close to 0. Although the LD (r^2) among eQTL was not as low as the LD (r^2) among these randomly selected SNVs, they were still mostly below 0.5 with a median of 0.19. Therefore, we concluded most of the eQTL were independent.

Additionally, 'LD values' language is used, recommend being explicit that these are R2 values in the figure itself and the legend.

We split the histogram and labeled r^2 explicitly.

- Fig 4b (line 330) – add p-values for correlation coefficients

We modified this point as suggested.

- Fig 5/line 378: It is a bit confusing that in the top plots, color and threshold are both related to significance, while in the bottom plots, the threshold is the 99th percentile but only the colored points are also statistically significant

The significance by p -values and the threshold by the 99th percentile of mediation estimates are two different things. In the bottom left plot, some gray dots are above the 99th percentile. However, all points can pass the 99th percentile and also be significant by p -values, as in the bottom right plot. If it's really confusing, we could remove the 99th percentile for mediation estimates and only describe it. Or remove points that are uninterpretable, though the 99th percentile was calculated by all values.

- Supp Fig 4 and the discussion of this – It is not entirely clear what the reader is meant to take away from this figure; it might be helpful to more thoroughly explain (are the hotspots different from their local environment? What do you/would you expect to see if so?) or to remove the analysis.

We used this analysis to check if hotspots harbored more diversity or were under balancing selection. But because we only used distant eQTL outside of divergent regions (Lee et al. 2021), which have been found to harbor more diversity and under balancing selection, we did not see the sign of balancing selection for hotspots.

We decided to remove the analysis as suggested.

Additionally, it's somewhat confusing that there isn't higher nucleotide diversity evident on the arms when historical recombination and SNV burden is greater on the arms (as this manuscript demonstrates in Table 1)

We used "cM" on the x-axis to be consistent with Fig. 3, the hotspot plot. The differences of diversity between arms and centers are clear if "Mb" was used on the x-axis.

- Supp Fig 7 – This fine mapping is impressive, but this is too much data for a reader to take in. Some related thoughts and questions in case helpful: Perhaps these could be displayed on a webpage where one locus at a time could be shown? Or narrow down the number of plots shown by some criteria? Also, more information in the legend would be helpful, for example: Is each transcript shown only once per hotspot? How are transcripts grouped when the same hotspot is broken into two candidate genes? Could multiple candidate genes be highlighted together (i.e., in different colors) for all the transcripts in cases where there is multiple to help the reader build confidence about how the candidates were selected (to show that the transcripts for one candidate have different relationship to it than to another candidate)?

In Supp Fig7, a hotspot could have several common candidate genes. For each candidate gene of each hotspot, we made a plot, including all the transcripts with distant eQTL in this hotspot and had this gene as a candidate in the fine mapping. So a transcript might appear in more than one panel in Supp Fig7. We tried other ways to show the plot, such as indicating multiple candidate genes together, and found the initial Supp Fig7 with multiple panels gave the clearest visualization. But we agree Supp Fig7 is massive. As mentioned above for the integration of the data to CeNDR, fine mapping plots of each eQTL would be provided as outputs in the GWA mapping results. So for Supp Fig7, after merging adjacent hotspots as answered above, we modified it and only showed fine mapping plots for 10 traits that had *hil-2* as a candidate gene and were enriched in certain gene classes. The information of common candidate genes for each hotspot are still in the Supplementary data 4. The fine mapping results for all the distant eQTL that were shown in Supp Fig7 will be publically available in the github repository for this paper.

- line 585 (and anywhere else only number of transcripts is mentioned) – would be helpful to also report N genes, as is done throughout most of the paper

We modified this point as suggested.

“Then, we filtered 26,043 reliably expressed transcripts of 16,238 genes by requiring at least five normalized counts in all the replicates of at least ten strains (Supplementary Fig. 1).”

- Clarity in methods. Overall, the methods section and introduction to methods in results are reasonably clear, but two things in particular would benefit from expansion.

1) eQTL methods and results use two statistical significance thresholds without explaining why one isn't chosen and used consistently; the methods hint at this but a brief note would be helpful - at least to this reader - especially given this is pretty prominent: Fig 2c is split into two based on significance threshold; similarly, Fig 5 has some panels with hits detected using one threshold and other panels using another, without any clear explanation this reader could easily find, and is using 2 thresholds for both GWA mapping and eQTL display.

For eQTL, we calculated both EIGEN 5% FDR threshold and BF 5% FDR threshold. We used EIGEN 5% FDR threshold to identify eQTL, which we used for other analyses throughout the paper except in Fig. 2C. We think it's informative to show eQTL that explained most of the variance also had higher significance. In the supplementary data, we provided information on whether an eQTL only passed the EIGEN 5% FDR threshold or also passed BF 5% FDR threshold.

For GWA mappings of organism-level traits in Fig. 5, we first applied the threshold used in the original studies to identify QTL. But some different isotype strains in the original studies were now considered as the same isotypes and were removed. We also used a different VCF in GWA mappings. In two studies, we did not recapitulate the QTL in the original studies that used the BF threshold, so we used the EIGEN threshold instead. We added the following sentence in the Methods.

“To summarize, EIGEN thresholds were used in GWA QTL identification for responses to arsenic³¹ (Fig. 5b), zinc³⁰ (Fig. 5c), etoposide³² (Fig. 5d), propionate³³ (Fig. 5e), abamectin³⁴ (Fig. 5f), and dauer formation in response to pheromone³⁵; BF thresholds were used in GWA QTL identification for response to albendazole (Supplementary Fig. 6), telomere length³⁶ (Fig. 5a), and lifetime fecundity³⁷ (Fig. 5g).”

2) Mediation analysis methods especially line 780 – a fuller explanation of the nested mediation analysis would be helpful: why did the first analysis give uninterpretable results that then needed to be re-estimated? If this is sensible mathematically as I imagine it is, it would be helpful to briefly explain why.

We have previously performed mediation analysis using gene expression and eQTL data from the N2xCB4856 recombinant inbred lines (Evans and Andersen, G3, 2020; Evans et al., PloS Genet., 2020). In these studies, we used the R package *mediation*, which estimates the total effect (the estimated effect of genotype on phenotype, ignoring expression), the mediation effect (the estimated effect of expression on phenotype), the proportion of mediation effect in total effect and *p*-values. This “proportion” should be non-negative and less than or equal to 1. In these previous studies, negative proportion values or those higher than 1 were classified as uninterpretable and dropped. In this study, because we have much more data and larger QTL regions of interest (GWA QTL can be quite large because of LD), we had much more potential mediator to test for each trait. We decided to correct *p*-values using the R package *MultiMed*, which only estimates *p*-values, adjusted *p*-values by permutation, and the mediation effect. We could not know whether the results by *MultiMed* were interpretable. So, we next used *mediation* to have a more comprehensive understanding of the total and mediation effects only on significant candidates by adjusted *p*-values in the results of *MultiMed*. Furthermore, the analysis by *mediation* is much slower than by *MultiMed*. By applying nested mediation analyses, the total processing time was reduced. We added these points in the Methods for mediation analysis.

“The R package *MultiMed* could calculate the mediation effects of multiple mediators and the adjusted *p*-values efficiently, but it does not provide estimates of the “total effect” (the estimated effect of genotype on phenotype, ignoring expression) or the “proportion” of mediation effect in the total effect. This “proportion” should be non-negative and less than or equal to 1. We have previously² used the R package *mediation* (version 4.5.0)³⁸ to estimate “total effect” and “proportion”. Mediators with negative proportion values or those higher than 1 were classified as uninterpretable and dropped. However, calculation using *mediation* is more time-consuming than using *MultiMed*. Therefore, we performed a second mediation analysis using the *mediate()* function in *mediation* only for significant mediators (adjusted *p* < 0.05 or mediation estimate greater than the 99th

percentile of the distribution of mediation estimates) in the results of *MultiMed*. Then, we filtered out mediators with uninterpretable results.”

- Reporting summary – sample size. Here the number of animals per plate is discussed, but I believe it would make more sense to discuss how the number of strains sequenced was chosen – these are the distinct samples.

The processes of animal growth and harvesting were done in the summer of 2017 during a large-scale GWA project on *C. elegans* variation to drugs. At that time, we had 249 isotype strains in the collection (CeNDR version 20170531). We aimed to test every isotype strain for drug response with at least three biological replicates and harvested synchronized worms in the control condition for RNA-seq. We did not get triplicates for each of the 249 strains. A few harvested samples showed low quality or quantity during RNA isolation or RNA library construction. We sequenced all the strains with at least two high quality replicates.

We added these points in the Reporting summary – sample size:

“The processes of animal growth and harvesting were done during a large-scale GWA project on *C. elegans* variation to drugs in 2017. At that time, we had 249 isotype strains in the collection (CeNDR version 20170531). We aimed to test every isotype strain for drug response with at least three biological replicates and harvested synchronized worms in the control condition for RNA-seq. To prepare animals for RNA extraction, we grew approximately 1,000 *C. elegans* embryos on each 10 cm plate to the young adult stage. This sample size avoids overcrowding and starvation, and guarantees enough total RNA for sequencing library construction. We did not get triplicates for each of the 249 strains. A few harvested samples showed low quality or quantity during RNA isolation or sequencing library construction. We performed RNA-seq on all the strains with at least two high-quality replicates.”

- Reporting summary/methods – it is not clear how strains were divided into batches or randomized, other than for RNA library construction where this is more thoroughly described in Methods. This should be clear at least in the reporting summary (were replicates from the same strain done on different days/different batches? How many total batches? etc)

In RNA isolation, replicates of the same strain were done in different batches. We did 12 to 24 samples per batch and 30 batches in total.

We added the following sentence in the Methods for RNA extraction.

“For the over 600 samples of 207 strains, we performed 30 batches of RNA extraction, with 12 to 24 samples per batch and replicates of the same strains in different batches.”

....Proofreading corrections and stylistic suggestions....

- Fig 2: consider including a legend describing colors in the figure itself especially for (a) which has different colors than the rest of the figure

We modified this point as suggested.

- Paragraph beginning line 476 is somewhat repetitive and has a different tone than the surrounding paragraphs; likely dispensable, or you could especially get rid of the last sentence.

We removed the last sentence as suggested.

- Recommend finding all the instances the adjective ‘diverse’ is used and replacing with synonyms at least some of the time, focusing on whether what is described can actually be ‘diverse’

Throughout the text, we replaced “diverse” with its synonyms and tried to use the diversity of terms correctly. Please see the document comparison to see specific changes.

- line 675 typo, %5 should be 5%

We modified this point as suggested.

- Line 806 acknowledgments, a couple missing/misplaced words, this is probably meant to read ‘We would also like to thank WormBase without which these analyses would not have been possible.’

We modified this point as suggested.

References:

1. Rockman, M. V., Skrovanek, S. S. & Kruglyak, L. Selection at linked sites shapes heritable phenotypic variation in *C. elegans*. *Science* **330**, 372–376 (2010).
2. Evans, K. S. & Andersen, E. C. The Gene *scb-1* Underlies Variation in *Caenorhabditis elegans* Chemotherapeutic Responses. *G3* **10**, 2353–2364 (2020).
3. Li, Y. *et al.* Mapping determinants of gene expression plasticity by genetical genomics in *C. elegans*. *PLoS Genet.* **2**, e222 (2006).
4. Viñuela, A., Snoek, L. B., Riksen, J. A. G. & Kammenga, J. E. Genome-wide gene expression regulation as a function of genotype and age in *C. elegans*. *Genome Res.* **20**, 929–937 (2010).
5. Li, Y. *et al.* Global genetic robustness of the alternative splicing machinery in *Caenorhabditis elegans*. *Genetics* **186**, 405–410 (2010).
6. Sterken, M. G. *et al.* Ras/MAPK Modifier Loci Revealed by eQTL in *Caenorhabditis elegans*. *G3* **7**, 3185–3193 (2017).
7. Snoek, B. L. *et al.* Contribution of trans regulatory eQTL to cryptic genetic variation in *C. elegans*. *BMC Genomics* **18**, 500 (2017).
8. Snoek, B. L. *et al.* The genetics of gene expression in a *Caenorhabditis elegans* multiparental recombinant inbred line population. *G3* **11**, (2021).
9. Ben-David, E. *et al.* Whole-organism eQTL mapping at cellular resolution with single-cell sequencing. *Elife* **10**, (2021).
10. Snoek, B. L. *et al.* WormQTL2: an interactive platform for systems genetics in *Caenorhabditis elegans*. *Database* **2020**, (2020).

11. Araya, C. L. *et al.* Regulatory analysis of the *C. elegans* genome with spatiotemporal resolution. *Nature* **512**, 400–405 (2014).
12. Kudron, M. M. *et al.* The ModERN Resource: Genome-Wide Binding Profiles for Hundreds of *Drosophila* and *Caenorhabditis elegans* Transcription Factors. *Genetics* **208**, 937–949 (2018).
13. Holdorf, A. D. *et al.* WormCat: An Online Tool for Annotation and Visualization of *Caenorhabditis elegans* Genome-Scale Data. *Genetics* **214**, 279–294 (2020).
14. Hahnel, S. R. *et al.* Extreme allelic heterogeneity at a *Caenorhabditis elegans* beta-tubulin locus explains natural resistance to benzimidazoles. *PLoS Pathog.* **14**, e1007226 (2018).
15. Dilks, C. M. *et al.* Quantitative benzimidazole resistance and fitness effects of parasitic nematode beta-tubulin alleles. *Int. J. Parasitol. Drugs Drug Resist.* **14**, 28–36 (2020).
16. Dilks, C. M., Koury, E. J., Buchanan, C. M. & Andersen, E. C. Newly identified parasitic nematode beta-tubulin alleles confer resistance to benzimidazoles. *Int. J. Parasitol. Drugs Drug Resist.* **17**, 168–175 (2021).
17. Cook, D. E., Zdraljevic, S., Roberts, J. P. & Andersen, E. C. CeNDR, the *Caenorhabditis elegans* natural diversity resource. *Nucleic Acids Res.* **45**, D650–D657 (2017).
18. Evans, K. S., van Wijk, M. H., McGrath, P. T., Andersen, E. C. & Sterken, M. G. From QTL to gene: *C. elegans* facilitates discoveries of the genetic mechanisms underlying natural variation. *Trends Genet.* **0**, (2021).
19. Widmayer, S. J., Evans, K., Zdraljevic, S. & Andersen, E. C. Evaluating the power and limitations of genome-wide association mapping in *C. elegans*. *bioRxiv* 2021.09.09.459688 (2021) doi:10.1101/2021.09.09.459688.
20. Bulteau, R. & Francesconi, M. Real age prediction from the transcriptome with RAPToR. *bioRxiv* 2021.09.07.459270 (2021) doi:10.1101/2021.09.07.459270.
21. Smith, E. N. & Kruglyak, L. Gene-environment interaction in yeast gene expression. *PLoS Biol.* **6**, e83 (2008).
22. Satterlee, J. S. *et al.* Specification of thermosensory neuron fate in *C. elegans* requires *ttx-1*, a homolog of *otd/Otx*. *Neuron* **31**, 943–956 (2001).

23. Kagoshima, H. & Kohara, Y. Co-expression of the transcription factors CEH-14 and TTX-1 regulates AFD neuron-specific genes *gcy-8* and *gcy-18* in *C. elegans*. *Dev. Biol.* **399**, 325–336 (2015).
24. Fairfax, B. P. *et al.* Innate immune activity conditions the effect of regulatory variants upon monocyte gene expression. *Science* **343**, 1246949 (2014).
25. Albert, F. W. & Kruglyak, L. The role of regulatory variation in complex traits and disease. *Nat. Rev. Genet.* **16**, 197–212 (2015).
26. Albert, F. W., Bloom, J. S., Siegel, J., Day, L. & Kruglyak, L. Genetics of trans-regulatory variation in gene expression. *Elife* **7**, 1–39 (2018).
27. Lee, D. *et al.* Balancing selection maintains hyper-divergent haplotypes in *Caenorhabditis elegans*. *Nat Ecol Evol* **5**, 794–807 (2021).
28. GTEx Consortium *et al.* Genetic effects on gene expression across human tissues. *Nature* **550**, 204–213 (2017).
29. GTEx Consortium. The GTEx Consortium atlas of genetic regulatory effects across human tissues. *Science* **369**, 1318–1330 (2020).
30. Evans, K. S. *et al.* Natural variation in the sequestosome-related gene, *sqst-5*, underlies zinc homeostasis in *Caenorhabditis elegans*. *PLoS Genet.* **16**, e1008986 (2020).
31. Zdraljevic, S. *et al.* Natural variation in *C. elegans* arsenic toxicity is explained by differences in branched chain amino acid metabolism. *Elife* **8**, e40260 (2019).
32. Zdraljevic, S. *et al.* Natural variation in a single amino acid substitution underlies physiological responses to topoisomerase II poisons. *PLoS Genet.* **13**, e1006891 (2017).
33. Na, H., Zdraljevic, S., Tanny, R. E., Walhout, A. J. M. & Andersen, E. C. Natural variation in a glucuronosyltransferase modulates propionate sensitivity in a *C. elegans* propionic acidemia model. *PLoS Genet.* **16**, e1008984 (2020).
34. Evans, K. S. *et al.* Two novel loci underlie natural differences in *Caenorhabditis elegans* abamectin responses. *PLoS Pathog.* **17**, e1009297 (2021).
35. Lee, D. *et al.* Selection and gene flow shape niche-associated variation in pheromone response. *Nat Ecol Evol* **3**, 1455–1463 (2019).

36. Cook, D. E. *et al.* The Genetic Basis of Natural Variation in *Caenorhabditis elegans* Telomere Length. *Genetics* **204**, 371–383 (2016).
37. Zhang, G., Mostad, J. D. & Andersen, E. C. Natural variation in fecundity is correlated with species-wide levels of divergence in *Caenorhabditis elegans*. *G3* (2021) doi:10.1093/g3journal/jkab168.
38. Tingley, D., Yamamoto, T., Hirose, K., Keele, L. & Imai, K. mediation: R Package for Causal Mediation Analysis. *Journal of Statistical Software, Articles* **59**, 1–38 (2014).

Reviewers' Comments:

Reviewer #1:

Remarks to the Author:

I'm happy with the responses.

Reviewer #2:

Remarks to the Author:

This is a compelling manuscript! The authors have done a thorough job of responding to the previous comments, resulting in a strengthened manuscript that will be of interest to the *C. elegans* and quantitative genetics communities. The changes to the title, fig 1d/e, eQTL hotspot sections, and mediation analysis description are especially well done and informative. Additionally, the details about the future CeNDR eQTL analysis are helpful to know (and exciting!).

Here are some thoughts for further – mostly small – improvements, which I include only in case helpful to the authors; I do not feel that I need to see another version of this manuscript before its publication. I look forward to reading the final, polished, published paper.

- On heritability (and reconciling differing definitions). It might be clarifying to parenthetically reference the key methods when initially discussing H^2 and h^2 (lines 139-140), for example say “broad sense heritability (H^2 , here calculated as strain-wise variance)” and similarly briefly note h^2 relates to the SNPs included in mapping in the main text.
 - On comparisons with previous eQTL studies (Reviewer 1’s main comments). I think these sections, currently in the discussion (paragraphs starting lines 396 and 419), would fit better in the results section, perhaps with both associated figures as supplementary figures. One option would be to include this around line 154 ‘in close agreement to previous *C. elegans* eQTL studies.’ I also highly recommend including the proportions of eQTLs overlapping vs. not across studies, especially highlighting the proportion newly discovered in this study, not just numbers, when describing these results to make it easier for the reader to interpret.
 - On thresholds (EIGEN, BF) – the clarifying wording in Methods helps the reader track the thresholds through the paper, but I still find it unclear why one threshold is chosen over another especially in Fig 2c. Perhaps clarify by noting in results how these thresholds are different? E.g. line 172, “the 2551 local eQTL that passed the Bonferroni 5% FDR threshold explained most of the estimated narrow-sense heritability”. – how are the thresholds different/used differently; why is the trend clear with the BF but not EIGEN thresholds. I may have just missed something simple here.
 - On the eQTL mapping being available in an upcoming CeNDR release, I really like the inclusion of this information in the manuscript. However, I’m not sure it should be the concluding paragraph of the entire paper. One possible place it might go better would be nearer the beginning of the discussion, near Line 384 which introduces the data generated here as a resource.
-Typos, grammar, stylistic.....
- I thought the earlier past tense version of the abstract was more standard than the updated present-tense version (‘we found’ more standard than ‘we find’) – but this is probably an editorial decision
 - Line 153 – “indicating major roles of additive genetic variation on expression variation than other genetic factors” – missing a word, perhaps still circular (can you better detect additive stuff?)
 - Fig 2d/Line 200 states $p < 2 \times 10^{-16}$. Recommend including exact p-value on plot rather than asterisks (I think exact p-values are editorial requirements, too – pull it from the p -value of the wilcox.test

results object).

- line 212 header might be stronger if it more directly summarized the findings presented in the section.

- line 384, 'an' should be 'a'

- The Tajima's D analysis was removed from the manuscript, but it's still referenced in discussion (line 447), probably makes sense to remove

REVIEWER COMMENTS

Reviewer #1 (Remarks to the Author):

I'm happy with the responses.

Reviewer #2 (Remarks to the Author):

This is a compelling manuscript! The authors have done a thorough job of responding to the previous comments, resulting in a strengthened manuscript that will be of interest to the *C. elegans* and quantitative genetics communities. The changes to the title, fig 1d/e, eQTL hotspot sections, and mediation analysis description are especially well done and informative. Additionally, the details about the future CeNDR eQTL analysis are helpful to know (and exciting!).

Here are some thoughts for further – mostly small – improvements, which I include only in case helpful to the authors; I do not feel that I need to see another version of this manuscript before its publication. I look forward to reading the final, polished, published paper.

- On heritability (and reconciling differing definitions). It might be clarifying to parenthetically reference the key methods when initially discussing H^2 and h^2 (lines 139-140), for example say “broad sense heritability (H^2 , here calculated as strain-wise variance)” and similarly briefly note h^2 relates to the SNPs included in mapping in the main text.

We clarified the definitions of heritability as suggested.

“To estimate the association between gene expression differences and genetic variation, we calculated the broad-sense heritability (H^2 , here calculated as strain-wise variance) and the narrow-sense heritability (h^2 , here calculated using the SNV matrix in the below GWA mappings) for each of the 25,849 transcript expression traits.”

- On comparisons with previous eQTL studies (Reviewer 1's main comments). I think these sections, currently in the discussion (paragraphs starting lines 396 and 419), would fit better in the results section, perhaps with both associated figures as supplementary figures. One option would be to include this around line 154 ‘in close agreement to previous *C. elegans* eQTL studies.’ I also highly recommend including the proportions of eQTLs overlapping vs. not across studies, especially highlighting the proportion newly discovered in this study, not just numbers, when describing these results to make it easier for the reader to interpret.

Overall, we disagree with the reviewer about the placement of these comparisons as results. In the recommended change (Line 154), the comparisons of hotspots would come before we present

our hotspot results. Perhaps the reviewer meant to add these comparisons after our presentation of the GWAS hotspot results? Even with this change, we feel that the flow from hotspots to mediation analyses would be disrupted. The current presentation of comparisons in the Discussion fits with the flow and does not detract from the mediation results.

However, we added details about which eQTL were found in our study uniquely and which eQTL were found in all eQTL studies.

- On thresholds (EIGEN, BF) – the clarifying wording in Methods helps the reader track the thresholds through the paper, but I still find it unclear why one threshold is chosen over another especially in Fig 2c. Perhaps clarify by noting in results how these thresholds are different? E.g. line 172, “the 2551 local eQTL that passed the Bonferroni 5% FDR threshold explained most of the estimated narrow-sense heritability”. – how are the thresholds different/used differently; why is the trend clear with the BF but not EIGEN thresholds. I may have just missed something simple here.

BF is a more stringent threshold than EIGEN. We used EIGEN to detect most possible eQTL and used BF to locate the best estimate of QTL positions. We mentioned this point in Methods, but in order to increase clarity, we modified the sentence on line 172 of the Results as:

“The 2,551 local eQTL that passed the more stringent Bonferroni 5% FDR threshold explained most of the estimated narrow-sense heritability (Fig. 2c).“

- On the eQTL mapping being available in an upcoming CeNDR release, I really like the inclusion of this information in the manuscript. However, I’m not sure it should be the concluding paragraph of the entire paper. One possible place it might go better would be nearer the beginning of the discussion, near Line 384 which introduces the data generated here as a resource.

We modified this point as suggested near Line 384.

“We used these data and GWA mappings to study gene regulation variation and developed a mediation analysis pipeline to identify causal candidates underlying variation in complex organism-level traits. In the latest version of CeNDR, we have implemented the mediation analysis using the expression and eQTL data into the GWA mapping tool¹ and provide the results along with other mapping outputs. We will further develop an expression browser on CeNDR for easy querying and interactive visualization of expression variation across wild strains. For genes and transcripts with detected eQTL, we will display the Manhattan plots and genotype-by-phenotype plots. We will also create tools to aid differential expression analysis and visualization between any pair of strains available in our data. We believe our data will further facilitate natural variation and evolutionary research in *C. elegans*.”

The last paragraph was removed.

.....Typos, grammar, stylistic.....

- I thought the earlier past tense version of the abstract was more standard than the updated present-tense version (‘we found’ more standard than ‘we find’) – but this is probably an editorial decision

Yes, the change to present-tense was according to formatting instructions.

- Line 153 – “indicating major roles of additive genetic variation on expression variation than other genetic factors” – missing a word, perhaps still circular (can you better detect additive stuff?)

We agree with the reviewer that this logic is circular and have removed the sentence.

- Fig 2d/Line 200 states $p < 2 \times 10^{-16}$. Recommend including exact p-value on plot rather than asterisks (I think exact p-values are editorial requirements, too – pull it from the \$p.value of the wilcox.test results object).

In R, the p-value is reported as $p < 2e-16$ when the value is very small and is not reported exactly (even from the wilcox.test object).

- line 212 header might be stronger if it more directly summarized the findings presented in the section.

We have changed the section header to:

“A diverse collection of molecular mechanisms underlie distant eQTL hotspots”

- line 384, ‘an’ should be ‘a’

We modified it as suggested.

- The Tajima’s D analysis was removed from the manuscript, but it’s still referenced in discussion (line 447), probably makes sense to remove

We removed the sentence as suggested. We also removed the calculation of Tajima’s D in Methods.

References

1. Widmayer, S. J., Evans, K., Zdraljevic, S. & Andersen, E. C. Evaluating the power and limitations of genome-wide association mapping in *C. elegans*. *bioRxiv* 2021.09.09.459688 (2021) doi:10.1101/2021.09.09.459688.